# On the equivalence between graph isomorphism testing and function approximation with GNNs

**Zhengdao Chen**
Courant Institute of Mathematical Sciences
New York University
zc1216@nyu.edu

**Soledad Villar**
Courant Institute of Mathematical Sciences
Center for Data Science
New York University
soledad.villar@nyu.edu

**Lei Chen**
Courant Institute of Mathematical Sciences
New York University
lc3909@nyu.edu

**Joan Bruna**
Courant Institute of Mathematical Sciences
Center for Data Science
New York University
bruna@cims.nyu.edu

## Abstract

Graph neural networks (GNNs) have achieved lots of success on graph-structured data. In light of this, there has been increasing interest in studying their representation power. One line of work focuses on the universal approximation of permutation-invariant functions by certain classes of GNNs, and another demonstrates the limitation of GNNs via graph isomorphism tests.

Our work connects these two perspectives and proves their equivalence. We further develop a framework of the representation power of GNNs with the language of sigma-algebra, which incorporates both viewpoints. Using this framework, we compare the expressive power of different classes of GNNs as well as other methods on graphs. In particular, we prove that order-2 Graph $G$-invariant networks fail to distinguish non-isomorphic regular graphs with the same degree. We then extend them to a new architecture, Ring-GNN, which succeeds in distinguishing these graphs as well as for tasks on real-world datasets.

## 1 Introduction

Graph structured data naturally occur in many areas of knowledge, including computational biology, chemistry and social sciences. Graph neural networks, in all their forms, yield useful representations of graph data partly because they take into consideration the intrinsic symmetries of graphs, such as invariance and equivariance with respect to a relabeling of the nodes [27, 7, 15, 8, 10, 28, 3, 36].

All these different architectures are proposed with different purposes (see [31] for a survey and references therein), and a priori it is not obvious how to compare their power. The recent work [32] proposes to study the representation power of GNNs via their performance on graph isomorphism tests. They developed the Graph Isomorphism Networks (GINs) that are as powerful as the one-dimensional Weisfeiler-Lehman (1-WL or just WL) test for graph isomorphism [30], and showed that no other neighborhood-aggregating (or message passing) GNN can be more powerful than the 1-WL test. Variants of message passing GNNs include [27, 9].

On the other hand, for feed-forward neural networks, many results have been obtained regarding their ability to approximate continuous functions, commonly known as the universal approximation theorems, such as the seminal works of [6, 12]. Following this line of work, it is natural to study

the expressivity of graph neural networks in terms of function approximation. Since we could argue that many if not most functions on a graph that we are interested in are invariant or equivariant to permutations of the nodes in the graph, GNNs are usually designed to be invariant or equivariant, and therefore the natural question is whether certain classes GNNs can approximate any continuous and invariant or equivariant functions. Recent work [19] showed the universal approximation of $G$-invariant networks, constructed based on the linear invariant and equivariant layers studied in [18], if the order of the tensor involved in the networks can grow as the graph gets larger. Such a dependence on the graph size was been theoretically overcame by the very recent work [13], though there is no known upper bound on the order of the tensors involved. With potentially very-high-order tensors, these models that are guaranteed of univeral approximation are not quite feasible in practice.

The foundational part of this work aims at building the bridge between graph isomorphism testing and invariant function approximation, the two main perspectives for studying the expressive power of graph neural networks. We demonstrate an equivalence between the the ability of a class of GNNs to distinguish between any pairs of non-isomorphic graph and its power of approximating any (continuous) invariant functions, for both the case with finite feature space and the case with continuous feature space. Furthermore, we argue that the concept of sigma-algebras on the space of graphs is a natural description of the power of graph neural networks, allowing us to build a taxonomy of GNNs based on how their respective sigmas-algebras interact. Building on this theoretical framework, we identify an opportunity to increase the expressive power of order-2 $G$-invariant networks with computational tractability, by considering a ring of invariant matrices under addition and multiplication. We show that the resulting model, which we refer to as *Ring-GNN*, is able to distinguish between non-isomorphic regular graphs where order-2 $G$-invariant networks provably fail. We illustrate these gains numerically in synthetic and real graph classification tasks.

Summary of main contributions:

- We show the equivalence between graph isomorphism testing and approximation of permutation-invariant functions for analyzing the expressive power of graph neural networks.

- We introduce a language of sigma algebra for studying the representation power of graph neural networks, which unifies both graph isomorphism testing and function approximation, and use this framework to compare the power of some GNNs and other methods.

- We propose Ring-GNN, a tractable extension of order-2 Graph $G$-invariant Networks that uses the ring of matrix addition and multiplication. We show this extension is necessary and sufficient to distinguish Circular Skip Links graphs.

## 2   Related work

**Graph Neural Networks and graph isomorphism.**   Graph isomorphism is a fundamental problem in theoretical computer science. It amounts to deciding, given two graphs $A$, $B$, whether there exists a permutation $\pi$ such that $\pi A = B\pi$. There exists no known polynomial-time algorithm to solve it, but recently Babai made a breakthrough by showing that it can be solved in quasi-polynomial-time [1]. Recently [32] introduced graph isomorphism tests as a characterization of the power of graph neural networks. They show that if a GNN follows a neighborhood aggregation scheme, then it cannot distinguish pairs of non-isomorphic graphs that the 1-WL test fails to distinguish. Therefore this class of GNNs is at most as powerful as the 1-WL test. They further propose the Graph Isomorphism Networks (GINs) based on approximating injective set functions by multi-layer perceptrons (MLPs), which can be as powerful as the 1-WL test. Based on $k$-WL tests [4], [20] proposes $k$-GNN, which can take higher-order interactions among nodes into account. Concurrently to this work, [17] proves that order-$k$ invariant graph networks are at least as powerful as the $k$-WL tests, and similarly to us, it and augments order-2 networks with matrix multiplication. They show they achieve at least the power of 3-WL test. [21] proposes relational pooling (RP), an approach that combines *permutation-sensitive* functions under all permutations to obtain a permutation-invariant function. If RP is combined with permutation-sensitive functions that are sufficiently expressive, then it can be shown to be a universal approximator. A combination of RP and GINs is able to distinguish certain non-isomorphic regular graphs which GIN alone would fail on. A drawback of RP is that its full version is intractable computationally, and therefore it needs to be approximated by averaging over randomly sampled permutations, in which case the resulting functions is not guaranteed to be permutation-invariant.

**Universal approximation of functions with symmetry.** Many works have discussed the function approximation capabilities of neural networks that satisfy certain symmetries. [2] studies the probablisitic and functional symmetry in neural networks, and we discuss its relationship to our work in more detail in Appendix D. [25] shows that equivariance of a neural network corresponds to symmetries in its parameter-sharing scheme. [35] proposes a neural network architecture with polynomial layers that is able to achieve universal approximation of invariant or equivariant functions. [18] studies the spaces of all invariant and equivariant linear functions, and obtained bases for such spaces. Building upon this work, [19] proposes the $G$-invariant network for a symmetry group $G$, which achieves universal approximation of $G$-invariant functions if the maximal tensor order involved in the network to grow as $\frac{n(n-1)}{2}$, but such high-order tensors are prohibitive in practice. Upper bounds on the approximation power of the $G$-invariant networks when the tensor order is limited remains open except for when $G = A_n$ [19]. The very recent work [13] extends the result to the equivariant case, although it suffers from the same problem of possibly requiring high-order tensors. Specifically for learning in graphs, [26] proposes the compositional networks, which achieve equivariance and are inspired by the WL test. In the context of machine perception of visual scenes, [11] proposes an architecture that can potentially express all equivariant functions.

To the best our knowledge, this is the first work that shows an explicit connection between the two aforementioned perspectives of studying the representation power of graph neural networks - graph isomorphism testing and universal approximation. Our main theoretical contribution lies in showing an equivalence between them, for both finite and continuous feature space cases, with a natural generalization of the notion of graph isomorphism testing to the latter case. Then we focus on the Graph $G$-invariant network based on [18, 19], and showed that when the maximum tensor order is restricted to be 2, then it cannot distinguish between non-isomorphic regular graphs with equal degrees. As a corollary, such networks are not universal. Note that our result shows an upper bound on order 2 $G$-invariant networks, whereas concurrently to us, [17] provides a lower bound by relating to $k$-WL tests. Concurrently to [17], we propose a modified version of order-2 graph networks to capture higher-order interactions among nodes without computing tensors of higher-order.

## 3 Graph isomorphism testing and universal approximation

In this section we show that there exists a very close connection between the universal approximation of permutation-invariant functions by a class of functions, and its ability to perform graph isomorphism tests. We consider graphs with nodes and edges labeled by elements of a compact set $\mathcal{X} \subset \mathbb{R}$. We represent graphs with $n$ nodes by an $n$ by $n$ matrix $G \in \mathcal{X}^{n \times n}$, where a diagonal term $G_{ii}$ represents the label of the $i$th node, and a non-diagonal $G_{ij}$ represents the label of the edge from the $i$th node to the $j$th node. An undirected graph will then be represented by a symmetric $G$.

Thus, we focus on analyzing a collection $\mathcal{C}$ of functions from $\mathcal{X}^{n \times n}$ to $\mathbb{R}$. We are especially interested in collections of *permutation-invariant functions*, defined so that $f(\pi^\intercal G \pi) = f(G)$, for all $G \in \mathcal{X}^{n \times n}$, and all $\pi \in S_n$, where $S_n$ is the permutation group of $n$ elements. For classes of functions, we define the property of being able to discriminate non-isomorphic graphs, which we call *GIso-discriminating*, which as we will see generalizes naturally to the continuous case.

**Definition 1.** *Let $\mathcal{C}$ be a collection of permutation-invariant functions from $\mathcal{X}^{n \times n}$ to $\mathbb{R}$. We say $\mathcal{C}$ is **GIso-discriminating** if for all non-isomorphic $G_1, G_2 \in \mathcal{X}^{n \times n}$ (denoted $G_1 \not\simeq G_2$), there exists a function $h \in \mathcal{C}$ such that $h(G_1) \neq h(G_2)$. This definition is illustrated by figure 2 in the appendix.*

**Definition 2.** *Let $\mathcal{C}$ be a collection of permutation-invariant functions from $\mathcal{X}^{n \times n}$ to $\mathbb{R}$. We say $\mathcal{C}$ is **universally approximating** if for all permutation-invariant function $f$ from $\mathcal{X}^{n \times n}$ to $\mathbb{R}$, and for all $\epsilon > 0$, there exists $h_{f,\epsilon} \in \mathcal{C}$ such that $\|f - h_{f,\epsilon}\|_\infty := \sup_{G \in \mathcal{X}^{n \times n}} |f(G) - h(G)| < \epsilon$*

### 3.1 Finite feature space

As a warm-up we first consider the space of graphs with a finite set of possible features for nodes and edges, $\mathcal{X} = \{1, \dots, M\}$.

**Theorem 1.** *Universally approximating classes of functions are also GIso-discriminating.*

*Proof.* Given $G_1, G_2 \in \mathcal{X}^{n \times n}$, we consider the permutation-invariant function $\mathbb{1}_{\simeq G_1} : \mathcal{X}^{n \times n} \to \mathbb{R}$ such that $\mathbb{1}_{\simeq G_1}(G) = 1$ if $G$ is isomorphic to $G_1$ and 0 otherwise. Therefore, it can be approximated

with $\epsilon = 0.1$ by a function $h \in \mathcal{C}$. Then $h$ is a function that distinguishes $G_1$ from $G_2$, as in Definition 1. Hence $\mathcal{C}$ is GIso-discriminating. □

To obtain a result on the reverse direction, we first introduce the concept of an augmented collection of functions, which is especially natural when $\mathcal{C}$ is a collection of neural networks.

**Definition 3.** *Given $\mathcal{C}$, a collection of functions from $\mathcal{X}^{n \times n}$ to $\mathbb{R}$, we consider an augmented collection of functions also from $\mathcal{X}^{n \times n}$ to $\mathbb{R}$ consisting of functions that map an input graph $G$ to $\mathcal{NN}([h_1(G), ..., h_d(G)])$ for some finite $d$, where $\mathcal{NN}$ is a feed-forward neural network / multi-layer perceptron, and $h_1, ..., h_d \in \mathcal{C}$. When $\mathcal{NN}$ is restricted to have $L$ layers, we denoted this augmented collection by $\mathcal{C}^{+L}$. In this work, we consider ReLU as the nonlinear activation function in the neural networks.*

**Remark 1.** *If $\mathcal{C}_{L_0}$ is the collection of feed-forward neural networks with $L_0$ layers, then $\mathcal{C}_{L_0}^{+L}$ represents the collection of feed-forward neural networks with $L_0 + L$ layers.*

**Remark 2.** *If $\mathcal{C}$ is a collection of permutation-invariant functions, so is $\mathcal{C}^{+L}$.*

**Theorem 2.** *If $\mathcal{C}$ is GIso-discriminating, then $\mathcal{C}^{+2}$ is universal approximating.*

The proof is simple and it is a consequence of the following lemmas that we prove in Appendix A.

**Lemma 1.** *If $\mathcal{C}$ is GIso-discriminating, then for all $G \in \mathcal{X}^{n \times n}$, there exists a function $\tilde{h}_G \in \mathcal{C}^{+1}$ such that for all $G', \tilde{h}_G(G') = 0$ if and only if $G \simeq G'$.*

**Lemma 2.** *Let $\mathcal{C}$ be a class of permutation-invariant functions from $\mathcal{X}^{n \times n}$ to $\mathbb{R}$ satisfying the consequences of Lemma 1, then $\mathcal{C}^{+1}$ is universally approximating.*

### 3.2 Extension to the case of continuous (Euclidean) feature space

Graph isomorphism is an inherently discrete problem, whereas universal approximation is usually more interesting when the input space is continuous. With our definition 1 of *GIso-discriminating*, we can achieve a natural generalization of the above results to the scenarios of continuous input space. All proofs for this section can be found in Appendix A.

Let $\mathcal{X}$ be a compact subset of $\mathbb{R}$, and we consider graphs with $n$ nodes represented by $G \in K = \mathcal{X}^{n \times n}$; that is, the node features are $\{G_{ii}\}_{i=1,...,n}$ and the edge features are $\{G_{ij}\}_{i,j=1,...,n;i \neq j}$.

**Theorem 3.** *If $\mathcal{C}$ is universally approximating, then it is also GIso-discriminating*

The essence of the proof is similar to that of Theorem 1. The other direction - showing that pairwise discrimination can lead to universal approximation - is less straightforward. As an intermediate step between, we make the following definition:

**Definition 4.** *Let $\mathcal{C}$ be a class of functions $K \to \mathbb{R}$. We say it is able to **locate every isomorphism class** if for all $G \in K$ and for all $\epsilon > 0$ there exists $h_G \in \mathcal{C}$ such that:*

- *for all $G' \in K, h_G(G') \geq 0$;*

- *for all $G' \in K$, if $G' \simeq G$, then $h_G(G') = 0$; and*

- *there exists $\delta_G > 0$ such that if $h_G < \delta_G$, then $\exists \pi \in S_n$ such that $d(\pi(G'), G) < \epsilon$, where $d$ is the Euclidean distance defined on $\mathbb{R}^{n \times n}$*

**Lemma 3.** *If $\mathcal{C}$, a collection of continuous permutation-invariant functions from $K$ to $\mathbb{R}$, is GIso-discriminating, then $\mathcal{C}^{+1}$ is able to locate every isomorphism class.*

Heuristically, we can think of the $h_G$ in the definition above as a "loss function" that penalizes the deviation of $G'$ from the equivalence class of $G$. In particular, the second condition says that if the loss value is small enough, then we know that $G'$ has to be close to the equivalence class of $G$.

**Lemma 4.** *Let $\mathcal{C}$ be a class of permutation-invariant functions $K \to \mathbb{R}$. If $\mathcal{C}$ is able to locate every isomorphism class, then $\mathcal{C}^{+2}$ is universally approximating.*

Combining the two lemmas above, we arrive at the following theorem:

**Theorem 4.** *If $\mathcal{C}$, a collection of continuous permutation-invariant functions from $K$ to $\mathbb{R}$, is GIso-discriminating, then $\mathcal{C}^{+3}$ is universaly approximating.*

# 4 A framework of representation power based on sigma-algebra

## 4.1 Introducing sigma-algebra to this context

Let $K = \mathcal{X}^{n \times n}$ be a finite input space. Let $Q_K := K/_{\simeq}$ be the set of isomorphism classes under the equivalence relation of graph isomorphism. That is, for all $\tau \in Q_K, \tau = \{\pi^\intercal G \pi : \pi \in \Gamma_n\}$ for some $G \in K$.

Intuitively, a maximally expressive collection of permutation-invariant functions, $\mathcal{C}$, will allow us to know exactly which isomorphism class $\tau$ a given graph $G$ belongs to, by looking at the outputs of certain functions in the collection applied to $G$. Heuristically, we can consider each function in $\mathcal{C}$ as a "measurement", which partitions that graph space $K$ according to the function value at each point. If $\mathcal{C}$ is powerful enough, then as a collection it will partition $K$ to be as fine as $Q_K$. If not, it is going to be coarser than $Q_K$. These intuitions motivate us to introduce the language of sigma-algebra.

Recall that an algebra on a set $K$ is a collection of subsets of $K$ that includes $K$ itself, is closed under complement, and is closed under finite union. Because $K$ is finite, we have that an algebra on $K$ is also a sigma-algebra on $K$, where a sigma-algebra further satisfies the condition of being closed under countable unions. Since $Q_K$ is a set of (non-intersecting) subsets of $K$, we can obtain the algebra generated by $Q_K$, defined as the smallest algebra that contains $Q_K$, and use $\sigma(Q_K)$ to denote the algebra (and sigma-algebra) generated by $Q_K$.

**Observation 1.** *If $f : \mathcal{X}^{n \times n} \to \mathbb{R}$ is a permutation-invariant function, then $f$ is measurable with respect to $\sigma(Q_K)$, and we denote this by $f \in \mathcal{M}[\sigma(Q_K)]$*

Now consider a class of functions $\mathcal{C}$ that is permutation-invariant. Then for all $f \in \mathcal{C}, f \in \mathcal{M}[\sigma(Q_K)]$. We define the sigma-algebra generated by $f$ as the set of all the pre-images of Borel sets on $\mathbb{R}$ under $f$, and denote it by $\sigma(f)$. It is the smallest sigma-algebra on $K$ that makes $f$ measurable. For a class of functions $\mathcal{C}$, $\sigma(\mathcal{C})$ is defined as the smallest sigma-algebra on $K$ that makes all functions in $\mathcal{C}$ measurable. Because we assume $K$ is finite, it does not matter whether $\mathcal{C}$ is a countable collection.

## 4.2 Reformulating graph isomorphism testing and universal approximation with sigma-algebra

We restrict our attention to finite feature space case. Given a graph $G \in \mathcal{X}^{n \times n}$, we use $\mathcal{E}(G)$ to denote its isomorphism class, $\{G' \in \mathcal{X}^{n \times n} : G' \simeq G\}$. We prove the following results in Appendix B.

**Theorem 5.** *If $\mathcal{C}$ is a class of permutation-invariant functions on $\mathcal{X}^{n \times n}$ and $\mathcal{C}$ is GIso-discriminating, then $\sigma(\mathcal{C}) = \sigma(Q_K)$*

Together with Theorem 1, the following is an immediate consequence:

**Corollary 1.** *If $\mathcal{C}$ is a class of permutation-invariant functions on $\mathcal{X}^{n \times n}$ and $\mathcal{C}$ achieves universal approximation, then $\sigma(\mathcal{C}) = \sigma(Q_K)$.*

**Theorem 6.** *Let be $\mathcal{C}$ a class of permutation-invariant functions on $\mathcal{X}^{n \times n}$ with $\sigma(\mathcal{C}) = \sigma(Q_K)$. Then $\mathcal{C}$ is GIso-discriminating.*

Thus, this sigma-algebra language is a natural notion for characterizing the power of graph neural networks, because as shown above, generating the finest sigma-algebra $\sigma(Q_K)$ is equivalent to being GIso-discriminating, and therefore to universal approximation.

Moreover, when $\mathcal{C}$ is not GIso-discriminating or universal, we can evaluate its representation power by studying $\sigma(\mathcal{C})$, which gives a measure for comparing the power of different GNN families. Given two classes of functions $\mathcal{C}_1, \mathcal{C}_2$, there is $\sigma(\mathcal{C}_1) \subseteq \sigma(\mathcal{C}_2)$ if and only if $\mathcal{M}[\sigma(\mathcal{C}_1)] \subseteq \mathcal{M}[\sigma(\mathcal{C}_2)]$ if and only if $\mathcal{C}_1$ is less powerful than $\mathcal{C}_2$ in terms of representation power. In Appendix C, we use this notion to compare the expressive power of different families of GNNs as well as other algorithms like 1-WL, linear programming and semidefinite programming in terms of their ability to distinguish non-isomorphic graphs. We summarize our findings in Figure 1.

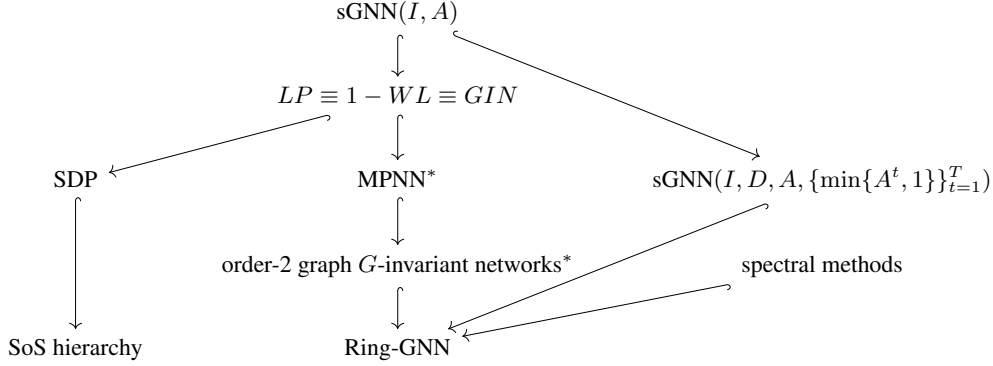

Figure 1: Comparison of classes of functions on graphs in terms of their expressive power under the sigma-algebra framework proposed in Section 4. *Remarks*: (a) GIN being defined in [32] as a form of message passing neural network (MPNN) justifies the inclusion GIN $\hookrightarrow$ MPNN. (b) [18] shows that message passing neural networks can be expressed as a modified form of order-2 graph $G$-invariant networks (which may not coincide with the definition we consider in this paper). Therefore this branch of the hierarchy has yet to be established rigorously. The rest of the figure is explained in Appendix C.

# 5 Ring-GNN: a GNN defined on the ring of equivariant functions

## 5.1 The limitation of order-2 Graph $G$-invariant Networks

We first investigate the $G$-invariant networks proposed in [19]. They are constructed by interleaving compositions of equivariant linear layers between tensors of potentially different orders and point-wise nonlinear activation functions. We define its adaptation to graph-structured data in Appendix E, and refer to it as *Graph $G$-invariant Networks*. It is a powerful framework that can achieve universal approximation if the order of the tensors can grow polynomially in the number of nodes [19], but less is known about its approximation power when the tensor order is restricted. One particularly interesting subclass of $G$-invariant networks is the ones with maximum tensor order 2 (we will call them *order-2 Graph $G$-invariant Networks*), because [18] shows that it can approximate any Message Passing Neural Network (MPNN) [8], and moreover, it would be both mathematically cumbersome and computationally expensive to include linear layers involving tensors with order higher than 2.

Our following result shows that the class of order-2 Graph $G$-invariant Networks is quite restrictive. The proof is given in Appendix E.

**Theorem 7.** *Order-2 Graph $G$-invariant Networks cannot distinguish between non-isomorphic regular graphs with the same degree.*

## 5.2 Ring-GNN as an extension of order-2 Graph $G$-invariant Networks

Motivated by this limitation, we propose a GNN architecture that extends the family of order-2 Graph $G$-invariant Networks without going into higher order tensors. In particular, we want the new family to include GNNs that can distinguish some pairs of non-isomorphic regular graphs with the same degree. For instance, take the pair of Circular Skip Link graphs $G_{8,2}$ and $G_{8,3}$, illustrated in Figure 5.2. Roughly speaking, if all the nodes in both graphs have the same node feature, then because they all have the same degree, the updates of node states in both graph neural networks based on neighborhood aggregation and the WL test will fail to distinguish the nodes. However, the *power graphs*[1] of $G_{8,2}$ and $G_{8,3}$ have different degrees. Another important example comes from spectral methods that operate on *normalized* operators, such as the normalized Laplacian $\Delta = I - D^{-1/2}AD^{-1/2}$, where $D$ is the diagonal degree operator. Such normalization preserves the permutation symmetries and in many clustering applications leads to dramatic improvements [29].

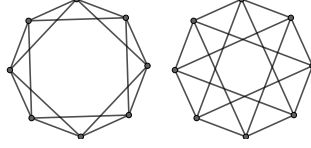

Figure 2: The Circular Skip Link graphs $G_{n,k}$ are undirected graphs in $n$ nodes $q_0, \ldots, q_{n-1}$ so that $(i,j) \in E$ if and only if $|i - j| \equiv 1$ or $k \pmod{n}$. In this figure we depict (left) $G_{8,2}$ and (right) $G_{8,3}$. It is very easy to check that $G_{n,k}$ and $G_{n',k'}$ are not isomorphic unless $n = n'$ and $k \equiv \pm k' \pmod{n}$. Both 1-WL and $G$-invariant networks fail to distinguish them.

This motivates us to consider a polynomial ring generated by the matrices that are the outputs of permutation-equivariant linear layers, rather than just the linear space of those outputs. Together with point-wise nonlinear activation functions such as ReLU, power graph adjacency matrices like $\min(A^2, 1)$ can be expressed with suitable choices of parameters.

To start with, we revisit the theory of linear equivariant functions developed in [18]. It is shown that any linear equivariant layer from $\mathbb{R}^{n \times n}$ to $\mathbb{R}^{n \times n}$ can be represented as $L_\theta(A) = \sum_{i=1}^{15} \theta_i L_i(A) + \sum_{i=16}^{17} \theta_i \overline{L}_i$, where $\{L_i\}_{i=1,\ldots,15}$ is the set of 15 basis functions for all linear equivariant functions from $\mathbb{R}^{n \times n}$ to $\mathbb{R}^{n \times n}$, $\overline{L}_{16}$ and $\overline{L}_{17}$ are the basis for the bias terms, and $\theta \in \mathbb{R}^{17}$ are the parameters that determine $L$. Generalizing to an equivariant linear layer from $\mathbb{R}^{n \times n \times d}$ to $\mathbb{R}^{n \times n \times d'}$, it becomes $L_\theta(A)_{\cdot,\cdot,k'} = \sum_{k=1}^{d} \sum_{i=1}^{15} \theta_{k,k',i} L_i(A_{\cdot,\cdot,i}) + \sum_{i=16}^{17} \theta_{k,k',i} \overline{L}_i$, with $\theta \in \mathbb{R}^{d \times d' \times 17}$.

With this in mind, we now define a new architecture. Suppose the input is $A^{(0)} \in \mathbb{R}^{n \times n \times d}$, containing data on a graph with $n$ nodes. We fix some integer $T$, and for $t = \{0, \ldots, T-1\}$, iteratively define

$$
\begin{aligned}
B_1^{(t)} &= \sigma(L_{\alpha^{(t)}}(A^{(t)})) \\
B_2^{(t)} &= \sigma(L_{\beta^{(t)}}(A^{(t)}) \cdot L_{\gamma^{(t)}}(A^{(t)})) \\
A^{(t+1)} &= k_1^{(t)} B_1^{(t)} + k_2^{(t)} B_2^{(t)}
\end{aligned}
$$

where $k_1^{(t)}, k_2^{(t)} \in \mathbb{R}$, $\alpha^{(t)}, \beta^{(t)}, \gamma^{(t)} \in \mathbb{R}^{d^{(t)} \times d'^{(t)} \times 17}$ are learnable parameters, and $\sigma$ is a pointwise nonlinear activation function such as ReLU. If a scalar output is desired, then in the final layer we compute $\theta_S \sum_{i,j} A_{ij}^{(T)} + \theta_D \sum_{i,i} A_{ii}^{(T)}$, where $\theta_S, \theta_D \in \mathbb{R}$ are trainable parameters. We call the resulting architecture the *Ring-GNN*.[2]

Note that each layer is equivariant, and the map from $A$ to the final scalar output is invariant. A Ring-GNN can reduce to an order-2 Graph $G$-invariant Network if $k_2^{(t)} = 0$. With $J + 1$ layers and suitable choices of the parameters, it is possible to obtain $\min(A^{2^J}, 1)$ in the $(J+1)^{th}$ layer. Therefore, we expect it to succeed in distinguishing certain pairs of regular graphs that order-2 Graph $G$-invariant Networks fail on, such as the Circular Skip Link graphs. Indeed, this is verified in the synthetic experiment presented in the next section. The normalized Laplacian can also be approximated, since the degree matrix can be inverted by taking the reciprocal on the diagonal, and then entry-wise inversion and square root on the diagonal can be approximated by MLPs.

Computationally, the complexity of running the forward model grows as $O(n^3)$, dominated by the matrix multiplications, which are what enable the computations of certain higher-order information as the depth increases. In comparison, a Graph $G$-invariant Network with maximal tensor order $k$ will have complexity at least $O(n^k)$. Therefore, the Ring-GNN is able to explore some higher-order interactions in the graph (which order-2 Graph $G$-invariant Networks neglect) while remaining computationally tractable. We note also that Ring-GNN can be augmented with matrix inverses or more generally with functional calculus on the spectrum of any of the intermediate representations[3] while keeping $O(n^3)$ computational complexity.

# 6 Experiments

The different models and the detailed setup of the experiments are discussed in Appendix F[4]. All experiments are conducted on GeForce GTX 1080 Ti and RTX 2080 Ti.

## 6.1 Classifying Circular Skip Links (CSL) graphs

The following experiment on synthetic data demonstrates the connection between function fitting and graph isomorphism testing. The Circular Skip Links graphs[5] are undirected regular graphs with node degree 4 [21], as illustrated in Figure 5.2. Note that two CSL graphs $G_{n,k}$ and $G_{n',k'}$ are not isomorphic unless $n = n'$ and $k \equiv \pm k' \pmod{n}$. In the experiment, which has the same setup as in [21], we fix $n = 41$, and set $k \in \{2, 3, 4, 5, 6, 9, 11, 12, 13, 16\}$, and each $k$ corresponds to a distinct isomorphism class. The task is then to classify a graph $G_{n,k}$ by its skip length $k$.

Note that since the 10 classes have the same size, a naive uniform classifier would obtain 10% accuracy. As we see from Table 1, both GIN and $G$-invariant network with tensor order 2 do not outperform the naive classifier. Their failure in this task is unsurprising: WL tests are proved to fall short of distinguishing such pairs of non-isomorphic regular graphs [4], and hence neither can GIN [32]; by the theoretical results from the previous section, order-2 Graph $G$-invariant network are unable to distinguish them either. Therefore, their failure as graph isomorphism tests is consistent with their failure in this classification task, which can be understood as trying to approximate the function that maps the graph to their class labels.

It should be noted that, since graph isomorphism tests are not entirely well-posed as classification tasks, the performance of GNN models could vary due to randomness. But the fact that Ring-GNNs achieve a relatively high maximum accuracy (compared to RP for example) demonstrates that as a class of GNNs it is rich enough to contain functions that distinguish the CSL graphs to a large extent.

| GNN architecture | Circular Skip Links | | | IMDBB | | IMDBM | |
|---|---|---|---|---|---|---|---|
| | max | min | std | mean | std | mean | std |
| RP-GIN † | 53.3 | 10 | 12.9 | - | - | - | - |
| GIN † ‡ | 10 | 10 | 0 | 75.1 | 5.1 | 52.3 | 2.8 |
| Order-2 Graph $G$-inv. † | 10 | 10 | 0 | 71.3 | 4.5 | 48.6 | 3.9 |
| sGNN-5 | 80 | 80 | 0 | 72.8 | 3.8 | 49.4 | 3.2 |
| sGNN-2 | 30 | 30 | 0 | 73.1 | 5.2 | 49.0 | 2.1 |
| sGNN-1 | 10 | 10 | 0 | 72.7 | 4.9 | 49.0 | 2.1 |
| LGNN [5] | 30 | 30 | 0 | 74.1 | 4.6 | 50.9 | 3.0 |
| Ring-GNN | 80 | 10 | 15.7 | 73.0 | 5.4 | 48.2 | 2.7 |
| Ring-GNN (w/ degree) ‡ | - | - | - | 73.3 | 4.9 | 51.3 | 4.2 |

Table 1: **(left)** Accuracy of different GNNs at classifying CSL (see Section 6.1). We report the best and worst performances among 10 experiments. **(right)** Accuracy of different GNNs at classifying real datasets IMDBB, IMDBM [34] (see Section 6.2). We report the best performance among all 350 epochs on 10-fold cross-validation, as was done in [32]. †: Reported performance by [21], [32] and [18]. ‡: On the IMDB datasets, unlike the other models, both GIN and the Ring-GNN (w/ degree) on the last row take the node degrees as input node features (see Section 6.2).

## 6.2 IMDB datasets

We use the two IMDB datasets (IMDBBINARY, IMDBMULTI)[6] [34] to test different models in real-world scenarios. Since our focus is on distinguishing graph structures, these datasets are suitable as they do not contain node features. IMDBBINARY has 1000 graphs, with average number of nodes 19.8 and 2 classes. IMDBMULTI has 1500 graphs, with average number of nodes 13.0 and 3 classes. Both datasets are randomly partitioned into $9:1$ for training/validation. As these two social network datasets have no informative node features, GIN uses one-hot encodings of node degrees as input node features, while the other baseline models treat all nodes to have identical features. For a fairer comparison, we apply two versions of Ring-GNN: the first one treats all nodes as having identical

input features and has identical depth and widths as the order-2 Graph $G$-invariant Network [18], denoted as "Ring-GNN" in Table 1; the second one uses the node degree as input features (though not as one-hot encodings, due to computational constraints, but simply as one integer per node), denoted as "Ring-GNN w/ degree" in Table 1. All models are evaluated via 10-fold cross validation and best accuracy is calculated through averaging across folds followed by maximizing along epochs [32]. Table 1 shows that Ring-GNN models achieve higher or similar performance compared to the order-2 Graph $G$-invariant networks on both datasets, and slightly worse performance compared to GIN.

### 6.3 Other real-world datasets

We perform further experiments on four other real-world datasets for classification tasks, including a social network dataset, COLLAB, and three bioinformatics datasets, MUTAG, PTC, PROTEINS[7] [34]. The experiment setup (10-fold cross validation, training/validation split) is identical to that of the IMDB datasets, except that all the bioinformatics datasets contain node features, and more details of hyperparameters are included in Appendix F. As shown in Table 2, Ring-GNN outperforms order-2 Graph $G$-invariant Network in all four datasets, and outperforms GIN in one out of the four datasets. Moreover, we note that the main goal of this part of our work is not necessarily to find the best-performing GNN through hyperparameter optimization, but rather to propose Ring-GNN as an augmented version of order-2 Graph $G$-invariant Networks and show experimental results that support the theory.

|  | COLLAB | MUTAG | PTC | PROTEINS |
|---|---|---|---|---|
| Ring-GNN | 80.1±1.4 | 86.8±6.4 | 65.7±7.1 | 75.7±2.9 |
| GIN † | 80.2±1.9 | 89.4±5.6 | 64.6±7.0 | 76.2±2.8 |
| Order-2 Graph $G$-inv. † | 77.9 ± 1.7 | 84.6±10.0 | 59.5±7.3 | 75.2±4.3 |

Table 2: Accuracy of different GNNs evaluated on several other real-world datasets. We report the best performance among all epochs on 10-fold cross-validation. †: Reported by [32] and [18].

## 7 Conclusions

In this work we address the important question of organizing the fast-growing zoo of GNN architectures in terms of what functions they can and cannot represent. We follow the approach via the graph isomorphism test, and show that is equivalent to the other perspective via function approximation. We leverage our graph isomorphism reduction to augment order-2 G-invariant nets with the ring of operators associated with matrix multiplication, which gives provable gains in expressive power with complexity $O(n^3)$, and is amenable to efficiency gains by leveraging sparsity in the graphs.

Our general framework leaves many interesting questions unresolved. First, a more comprehensive analysis on which elements of the algebra are really needed depending on the application. Next, our current GNN taxonomy is still incomplete, and in particular we believe it is important to further discern the abilities between spectral and neighborhood-aggregation-based architectures. Finally, and most importantly, our current notion of invariance (based on permutation symmetry) defines a topology in the space of graphs that is too strong; in other words, two graphs are either considered equal (if they are isomorphic) or not. Extending the theory of symmetric universal approximation to take into account a weaker metric in the space of graphs, such as the Gromov-Hausdorff distance, is a natural next step, that will better reflect the stability requirements of powerful graph representations to small graph perturbations in real-world applications.

**Acknowledgements** We would like to thank Haggai Maron and Thomas Kipf for fruitful discussions and for pointing us towards $G$-invariant networks as powerful models to study representational power in graphs. We thank Prof. Michael M. Bronstein for supporting this research with computing resources. This work was partially supported by NSF grant RI-IIS 1816753, NSF CAREER CIF 1845360, the Alfred P. Sloan Fellowship, Samsung GRP and Samsung Electronics. SV was partially funded by EOARD FA9550-18-1-7007 and the Simons Collaboration Algorithms and Geometry.

## Footnotes

[1]If $A$ is the adjacency matrix of a graph, its power graph has adjacency matrix $\min(A^2, 1)$. The matrix $\min(A^2, 1)$ has been used in [5] in graph neural networks for community detection and in [22] for the quadratic assignment problem, and it leverages multiscale information in the graph. Note that it differs from taking the power of certain matrices, which is exploited in [16] for example.

[2]We call it Ring-GNN since the main object we consider is the ring of matrices, but technically we can express an associative algebra since our model includes scalar multiplications.

[3]When $A = A^{(0)}$ is the adjacency matrix of an undirected graph, one easily verifies that $A^{(t)}$ contains only symmetric matrices for all $t$.

[4]The code is available at `https://github.com/leichen2018/Ring-GNN`.

[5]CSL dataset: `https://github.com/PurdueMINDS/RelationalPooling/tree/master/`

[6]IMDB datasets: `https://github.com/weihua916/powerful-gnns/blob/master/dataset.zip`

[7]Real datasets: `https://github.com/weihua916/powerful-gnns/blob/master/dataset.zip`
