[Supplementary Material]

# References

[1] László Babai. Graph isomorphism in quasipolynomial time. In *Proceedings of the forty-eighth annual ACM symposium on Theory of Computing*, pages 684–697. ACM, 2016.

[2] Benjamin Bloem-Reddy and Yee Whye Teh. Probabilistic symmetry and invariant neural networks. *arXiv preprint arXiv:1901.06082*, 2019.

[3] M. M. Bronstein, J. Bruna, Y. LeCun, A. Szlam, and P. Vandergheynst. Geometric deep learning: Going beyond euclidean data. *IEEE Signal Processing Magazine*, 34(4):18–42, July 2017.

[4] Jin-Yi Cai, Martin Fürer, and Neil Immerman. An optimal lower bound on the number of variables for graph identification. *Combinatorica*, 12(4):389–410, 1992.

[5] Zhengdao Chen, Lisha Li, and Joan Bruna. Supervised community detection with line graph neural networks. *Internation Conference on Learning Representations*, 2019.

[6] George Cybenko. Approximation by superpositions of a sigmoidal function. *Mathematics of control, signals and systems*, 2(4):303–314, 1989.

[7] David K Duvenaud, Dougal Maclaurin, Jorge Iparraguirre, Rafael Bombarell, Timothy Hirzel, Alán Aspuru-Guzik, and Ryan P Adams. Convolutional networks on graphs for learning molecular fingerprints. In *Advances in neural information processing systems*, pages 2224–2232, 2015.

[8] Justin Gilmer, Samuel S Schoenholz, Patrick F Riley, Oriol Vinyals, and George E Dahl. Neural message passing for quantum chemistry. In *Proceedings of the 34th International Conference on Machine Learning-Volume 70*, pages 1263–1272. JMLR. org, 2017.

[9] Will Hamilton, Zhitao Ying, and Jure Leskovec. Inductive representation learning on large graphs. In *Advances in Neural Information Processing Systems*, pages 1024–1034, 2017.

[10] William L Hamilton, Rex Ying, and Jure Leskovec. Representation learning on graphs: Methods and applications. *arXiv preprint arXiv:1709.05584*, 2017.

[11] Roei Herzig, Moshiko Raboh, Gal Chechik, Jonathan Berant, and Amir Globerson. Mapping images to scene graphs with permutation-invariant structured prediction. In *Advances in Neural Information Processing Systems*, pages 7211–7221, 2018.

[12] Kurt Hornik. Approximation capabilities of multilayer feedforward networks. *Neural Networks*, 4:251–257, 1991.

[13] Nicolas Keriven and Gabriel Peyré. Universal invariant and equivariant graph neural networks. *arXiv preprint arXiv:1905.04943*, 2019.

[14] Diederik P Kingma and Jimmy Ba. Adam: A method for stochastic optimization. *arXiv preprint arXiv:1412.6980*, 2014.

[15] Thomas N Kipf and Max Welling. Semi-supervised classification with graph convolutional networks. *arXiv preprint arXiv:1609.02907*, 2016.

[16] Renjie Liao, Zhizhen Zhao, Raquel Urtasun, and Richard Zemel. Lanczosnet: Multi-scale deep graph convolutional networks. In *International Conference on Learning Representations*, 2019.

[17] Haggai Maron, Heli Ben-Hamu, Hadar Serviansky, and Lipman Yaron. Provably powerful graph networks. *arXiv preprint arXiv:1905.11136*, 2019.

[18] Haggai Maron, Heli Ben-Hamu, Nadav Shamir, and Yaron Lipman. Invariant and equivariant graph networks. 2018.

[19] Haggai Maron, Ethan Fetaya, Nimrod Segol, and Yaron Lipman. On the universality of invariant networks. *arXiv preprint arXiv:1901.09342*, 2019.

[20] Christopher Morris, Martin Ritzert, Matthias Fey, William L Hamilton, Jan Eric Lenssen, Gaurav Rattan, and Martin Grohe. Weisfeiler and leman go neural: Higher-order graph neural networks. *Association for the Advancement of Artificial Intelligence*, 2019.

[21] Ryan L Murphy, Balasubramaniam Srinivasan, Vinayak Rao, and Bruno Ribeiro. Relational pooling for graph representations. *arXiv preprint arXiv:1903.02541*, 2019.

[22] Alex Nowak, Soledad Villar, Afonso S Bandeira, and Joan Bruna. A note on learning algorithms for quadratic assignment with graph neural networks. *arXiv preprint arXiv:1706.07450*, 2017.

[23] Ryan O'Donnell, John Wright, Chenggang Wu, and Yuan Zhou. Hardness of robust graph isomorphism, lasserre gaps, and asymmetry of random graphs. In *Proceedings of the twenty-fifth annual ACM-SIAM symposium on Discrete algorithms*, pages 1659–1677. Society for Industrial and Applied Mathematics, 2014.

[24] Motakuri V Ramana, Edward R Scheinerman, and Daniel Ullman. Fractional isomorphism of graphs. *Discrete Mathematics*, 132(1-3):247–265, 1994.

[25] Siamak Ravanbakhsh, Jeff Schneider, and Barnabas Poczos. Equivariance through parameter-sharing. *Proceedings of the 34th International Conference on Machine Learning*, 2017.

[26] Horace Pan Shubhendu Trivedi Brandon Anderson Risi Kondor, Hy Truong Son. Covariant compositional networks for learning graphs, 2018.

[27] Franco Scarselli, Marco Gori, Ah Chung Tsoi, Markus Hagenbuchner, and Gabriele Monfardini. The graph neural network model. *IEEE Transactions on Neural Networks*, 20(1):61–80, 2008.

[28] Petar Veličković, Guillem Cucurull, Arantxa Casanova, Adriana Romero, Pietro Lio, and Yoshua Bengio. Graph attention networks. *arXiv preprint arXiv:1710.10903*, 2017.

[29] Ulrike Von Luxburg. A tutorial on spectral clustering. *Statistics and computing*, 17(4):395–416, 2007.

[30] B Weisfeiler and A Leman. The reduction of a graph to canonical form and the algebra which appears therein. *Nauchno-Technicheskaya Informatsia*, 2(9):12-16, 1968.

[31] Zonghan Wu, Shirui Pan, Fengwen Chen, Guodong Long, Chengqi Zhang, and Philip S Yu. A comprehensive survey on graph neural networks. *arXiv preprint arXiv:1901.00596*, 2019.

[32] Keyulu Xu, Weihua Hu, Jure Leskovec, and Stefanie Jegelka. How powerful are graph neural networks? *arXiv preprint arXiv:1810.00826*, 2018.

[33] Keyulu Xu, Chengtao Li, Yonglong Tian, Tomohiro Sonobe, Ken-ichi Kawarabayashi, and Stefanie Jegelka. Representation learning on graphs with jumping knowledge networks. *arXiv preprint arXiv:1806.03536*, 2018.

[34] Pinar Yanardag and SVN Vishwanathan. Deep graph kernels. In *Proceedings of the 21th ACM SIGKDD International Conference on Knowledge Discovery and Data Mining*, pages 1365–1374. ACM, 2015.

[35] Dmitry Yarotsky. Universal approximations of invariant maps by neural networks. *arXiv preprint arXiv:1804.10306*, 2018.

[36] Jiaxuan You, Rex Ying, and Jure Leskovec. Position-aware graph neural networks. In Kamalika Chaudhuri and Ruslan Salakhutdinov, editors, *Proceedings of the 36th International Conference on Machine Learning*, pages 7134–7143, 2019.

[37] Qing Zhao, Stefan E Karisch, Franz Rendl, and Henry Wolkowicz. Semidefinite programming relaxations for the quadratic assignment problem. *Journal of Combinatorial Optimization*, 2(1):71–109, 1998.

# A Proofs on universal approximation and graph isomorphism

**Lemma 1.** If $\mathcal{C}$ is GIso-discriminating, then for all $G \in \mathcal{X}^{n \times n}$, there exists a function $\tilde{h}_G \in \mathcal{C}^{+1}$ such that for all $G', \tilde{h}_G(G') = 0$ if and only if $G \simeq G'$.

*Proof of Lemma 1.* Given $G, G' \in \mathcal{X}^{n \times n}$ with $G \not\simeq G'$, let $h_{G,G'} \in \mathcal{C}$ be the function that distinguishes this pair, i.e. $h_{G,G'}(G) \neq h_{G,G'}(G')$. Then define a function $\overline{h}_{G,G'}$ by

$$\begin{aligned}
\overline{h}_{G,G'}(G^*) &= |h_{G,G'}(G^*) - h_{G,G'}(G)| \\
&= \max(h_{G,G'}(G^*) - h_{G,G'}(G), 0) + \max(h_{G,G'}(G) - h_{G,G'}(G^*), 0)
\end{aligned} \tag{1}$$

Note that if $G^* \simeq G$, then $h_{G,G'}(G^*) = h_{G,G'}(G)$, and so $\overline{h}_{G,G'}(G^*) = 0$. If $G^* \simeq G'$, then $\overline{h}_{G,G'}(G^*) > 0$. Otherwise, $\overline{h}_{G,G'}(G^*) \geq 0$.

Next, define a function $\tilde{h}_G$ by $\tilde{h}_G(G^*) = \sum_{G' \in \mathcal{X}^{n \times n}, G' \not\simeq G} \overline{h}_{G,G'}(G^*)$. If $G^* \simeq G$, we have $\tilde{h}_G(G^*) = 0$, whereas if $G^* \not\simeq G$ then $\tilde{h}_G(G^*) > 0$.

Thus, it suffices to show that $\tilde{h}_G \in \mathcal{C}^{+1}$. We take the finite subcollection of functions, $\{h_{G,G'}\}_{G' \in \mathcal{X}^{n \times n}, G \not\simeq G'}$, and feed the input graph $G'$ to each of them to obtain a vector of outputs. By equation 1, $\overline{h}_{G,G'}(G^*)$ can be obtained from $h_{G,G'}(G^*)$ by passing through one ReLU layer. Finally, a finite summation across $G' \not\simeq G$ yields $\tilde{h}_G(G^*)$. Therefore, $\tilde{h}_G \in \mathcal{C}^{+1}, \forall G \in \mathcal{X}^{n \times n}$. $\square$

**Lemma 2** Let $\mathcal{C}$ be a class of permutation-invariant functions from $\mathcal{X}^{n \times n}$ to $\mathbb{R}$ so that for all $G \in \mathcal{X}^{n \times n}$, there exists $\tilde{h}_G \in \mathcal{C}$ satisfying $\tilde{h}_G(G') = 0$ if and only if $G \simeq G'$. Then $\mathcal{C}^{+1}$ is universally approximating.

*Proof of Lemma 2.* In fact, in the finite feature setting we can obtain a stronger result: for all $f$ that is permutation-invariant, $f \in \mathcal{C}^{+1}$, and so no approximation is needed.

We first use the $\tilde{h}_G$'s to construct all the indicator functions $\mathbb{1}_{G \simeq G^*}$ as functions of $G^*$ on $\mathcal{X}^{n \times n}$. To achieve this, because $\mathcal{X}^{n \times n}$ is finite, $\forall G$, we let $\delta_G = \frac{1}{2} \min_{G' \in \mathcal{X}^{n \times n}, G' \not\simeq G} |\tilde{h}_G(G')| > 0$. We then introduce a "bump" function from $\mathbb{R}$ to $\mathbb{R}$ with parameters $a$ and $b$, $\psi_{a,b}(x) = \psi((x - b)/a)$, where $\psi(x) = \max(x - 1, 0) + \max(x + 1, 0) - 2\max(x, 0)$. Then $\psi_{a,b}(b) = 0$, and $supp(\psi_{a,b}) = (b - a, b + a)$. Now, we define a function $\varphi_G$ from $\mathcal{X} = \{1, ..., M\}$ to $\mathbb{R}$ by $\varphi_G(G^*) = \psi_{\delta_G, 0}(\tilde{h}_G(G^*))$. Note that $\varphi_G(G^*) = \mathbb{1}_{G \simeq G^*}$ as a function of $G^*$ on $\mathcal{X}^{n \times n}$.

Given $f$, thanks to the finiteness of the input space $\mathcal{X}^{n \times n}$, we decompose it as $f(G^*) = (\frac{1}{|S_n|} \sum_{G \in \mathcal{X}^{n \times n}} \mathbb{1}_{G \simeq G^*}) f(G^*) = \frac{1}{|S_n|} \sum_{G \in \mathcal{X}^{n \times n}} f(G) \mathbb{1}_{G \simeq G^*} = \frac{1}{|S_n|} \sum_{G \in \mathcal{X}^{n \times n}} f(G) \varphi_G(G^*)$.

The right hand side can be realized in $\mathcal{C}^{+1}$, since we can first take the finite collection of functions $\{\tilde{h}_G\}_{G \in \mathcal{X}^{n \times n}}$ and obtain $\{\tilde{h}_G(G^*)\}_{G \in \mathcal{X}^{n \times n}}$. Then, with an MLP with one hidden layer, we can obtain $\{\varphi_G(G^*)\}_{G \in \mathcal{X}^{n \times n}}$, a linear combination of which gives the right hand side, since each "$f(G)$" within the summation is a constant. $\square$

**Theorem 3**. If $\mathcal{C}$ is universally approximating, then it is also GIso-discriminating

*Proof of Theorem 3.* $\forall G_1, G_2 \in K$, if $G_1 \not\simeq G_2$, define $f_1(G) = \min_{\pi \in S_n} d(G_1, \pi^\intercal G \pi)$. It is a continuous and permutation-invariant function on $K$, and therefore can be approximated by a function $h \in \mathcal{C}$ to within $\epsilon = \frac{1}{2} f_1(G_2) > 0$ accuracy. Then $h$ is a function that can discriminate between $G_1$ and $G_2$. $\square$

**Lemma 3**. If $\mathcal{C}$, a collection of continuous permutation-invariant functions from $K$ to $\mathbb{R}$, is pairwise distinguishing, then $\mathcal{C}^{+1}$ is able to locate every isomorphism class.

*Proof of Lemma 3.* Fix any $G \in K$. $\forall G' \not\simeq G \in K, \exists h_{G,G'} \in \mathcal{C}$ such that $h_{G,G'}(G) \neq h_{G,G'}(G')$. For each $G'$, define a set $A_{G'}$ as $h_{G,G'}^{-1}((h_{G,G'}(G') - \frac{|h_{G,G'}(G') - h_{G,G'}(G)|}{2}, h_{G,G'}(G') +$

Figure 3: Illustrating the definition of GIso-discriminating. $G, G'$ and $G''$ are mutually non-isomorphic, and each of the big circles with dashed boundary represents an equivalence class under graph isomorphism. $h_{G,G'}$ is a permutation-invariant function that obtains different values on equivalence class of $G$ and on that of $G'$, and similar $h_{G,G''}$. If the graph space has only these three equivalence classes of graphs, then $\mathcal{C} = \{h_{G,G'}, h_{G,G''}\}$ is GIso-discriminating.

$\frac{|h_{G,G'}(G') - h_{G,G'}(G)|}{2})) \subseteq K$. Obviously $G' \in A_{G'}$ and $G$ does not. Since $h_{G,G'}$ is assumed continuous, $A'_G$ is an open set for each $G' \not\simeq G$. If $G' \simeq G$, define $A_{G'} = B(G', \epsilon)$, the open $\epsilon$-ball in $K$ under the Euclidean distance.

Thus, $\{A_{G'}\}_{G' \in K}$ is an open cover of $K$. Since $K$ is compact, $\exists$ a finite subset $K_0$ of $K$ such that $\{A_{G'}\}_{G' \in K_0}$ also covers $K$.

Hence, $\forall G^* \in K, \exists G' \in K_0$ such that $G^* \in A_{G'}$. Moreover, $\forall G^* \in K \setminus (\bigcup_{G' \in \mathcal{E}(G)} A_{G'}) = K \setminus (\bigcup_{\pi \in S_n} B(\pi^\intercal G \pi, \epsilon))$, where $\mathcal{E}(G)$ represents the equivalence class of graphs in $K$ consisting of graphs isomorphic to $G$, $\exists G' \in K_0 \setminus \mathcal{E}(G)$ such that $G^* \in A_{G'}$.

Now define a function $\tilde{h}_G$ on $K$ by $\tilde{h}_G(G^*) = \sum_{G' \in K_0 \setminus \mathcal{E}(G)} \overline{h}_{G,G'}(G^*)$, where $\overline{h}_{G,G'}(G^*) = \max(\frac{2}{3}|h_{G,G'}(G) - h_{G,G'}(G')| - |h_{G,G'}(G^*) - h_{G,G'}(G')|, 0)$. Since each $h_{G,G'}$ in continuous, $\tilde{h}_G$ is also continuous. Thus, we can show that $\tilde{h}_G$ is the desired function in Definition 4:

- $h_{G,G'}$ is nonnegative $\forall G, G'$, and hence $\tilde{h}_G$ is nonnegative on $K$

- If $G^* \simeq G$, then as each $h_{G,G'}$ is permutation invariant, there is $h_{G,G'}(G^*) = h_{G,G'}(G)$, and hence $\overline{h}_{G,G'}(G^*) = 0$. Thus, $\tilde{h}_G(G^*) = 0$.

- If $\forall \pi \in S_n, d(\pi^\intercal G^* \pi, G) \geq \epsilon$, then $G^* \in K \setminus \bigcup_{G' \in \mathcal{E}(G)} A_{G'}$. Therefore, $\exists G' \in K \setminus \mathcal{E}(G)$ such that $G^* \in A_{G'}$, which implies that $|h_{G,G'}(G^*) - h_{G,G'}(G')| < \frac{1}{2}|h_{G,G'}(G) - h_{G,G'}(G')| < \frac{2}{3}|h_{G,G'}(G) - h_{G,G'}(G')|$. Therefore, $\frac{2}{3}|h_{G,G'}(G) - h_{G,G'}(G')| - |h_{G,G'}(G^*) - h_{G,G'}(G')| > \frac{1}{6}|h_{G,G'}(G) - h_{G,G'}(G')| > 0$, and so $\tilde{h}_G(G^*) \geq \overline{h}_{G,G'}(G^*) > \frac{1}{6}|h_{G,G'}(G) - h_{G,G'}(G')|$. Define $\delta_G = \frac{1}{6} \min_{G' \in K_0 \setminus \mathcal{E}(G)} |h_{G,G'}(G) - h_{G,G'}(G')| > 0$. Then if $\tilde{h}_G(G^*) < \delta_G$, it has to be the case that $G^* \in \bigcup_{G' \in \mathcal{E}(G)} A_{G'} = \bigcup_{\pi \in S_n} B(\pi^\intercal G \pi, \epsilon)$, implying that $\exists \pi \in S_n$ such that $d(G^*, \pi^\intercal G \pi) < \epsilon$.

Finally, it is clear that $\tilde{h}_G$ can be realized in $\mathcal{C}^{+1}$.

$\square$

**Lemma 4.** Let $\mathcal{C}$ be a class of permutation-invariant functions $K \to \mathbb{R}$. If $\mathcal{C}$ is able to locate every isomorphism class, then $\mathcal{C}^{+2}$ is universally approximating.

*Proof of Lemma 4.* Consider any $f$ that is continuous and permutation-invariant. Since $K$ is compact, $f$ is uniformly continuous on $K$. Therefore, $\forall \epsilon > 0, \exists r > 0$ such that $\forall G_1, G_2 \in K$, if $d(G_1, G_2) < r$, then $|f(G_1) - f(G_2)| < \epsilon$.

Given $\forall G \in K$, choose the function $h_G$ in definition 2. Use $h_G^{-1}(a)$ to denote $h_G^{-1}([0,a))$. Then $\exists \delta_G$ such that $h_G^{-1}(\delta_G) \subseteq B(G, r)$, where $B(G, r)$ is the ball in $K$ centered at $G$ with radius $r$ (in Euclidean distance). Since $h_G$ is continuous, $h_G^{-1}(\delta_G)$ is open. Therefore, $\{h_G^{-1}(\delta_G)\}_{G \in K}$ is an open cover of $K$. Because $K$ is compact, $\exists$ a finite subset $K_0 \subseteq K$ such that $\{h_G^{-1}(\delta_G)\}_{G \in K_0}$ also covers $K$.

$\forall G_0 \in K_0$, define another function $\varphi_{G_0}(G') = \delta_{G_0} - h_{G_0}(G')$ if $h_{G_0}(G') < \delta_{G_0}$ and 0 otherwise. Therefore, $\text{supp}(\varphi_{G_0}) = h_{G_0}^{-1}(\delta_{G_0})$. Let $\varphi(G') = \sum_{G^* \in K_0} \varphi_{G^*}(G')$, and then define $\psi_{G_0}(G') = \frac{\varphi_{G_0}(G')}{\varphi(G')}$. Note that $\forall G' \in K$, since $\{h_G^{-1}(\delta_G)\}_{G \in K_0}$ covers $K$, $\exists G^* \in K_0$ such that $G' \in h_{G^*}^{-1}(\delta_{G^*}) = \text{supp}(\varphi_{G^*})$, and so the denominator $> 0$. Therefore, $\psi_{G_0}$ is well defined on $K$, and $\text{supp}(\psi_{G_0}) = \text{supp}(\varphi_{G_0}) = h_{G_0}^{-1}(\delta_{G_0})$. Moreover, $\forall G' \in K, \sum_{G_0 \in K_0} \psi_{G_0}(G') = 1$. Therefore, the set of functions $\{\psi_{G_0}\}_{G_0 \in K_0}$ is a "partition of unity", with respect to the open cover $\{h_G^{-1}(\delta_G)\}_{G \in K_0}$.

Back to the function $f$ that we want to approximate. We want to express it in away that resembles what a neural network can do. With the set of functions $\{\psi_{G_0}\}_{G_0 \in K_0}$, we have

$$f(G') = \sum_{G_0 \in K_0} f(G')\psi_{G_0}(G') = \sum_{\substack{G_0 \in K_0 \\ G' \in h_{G_0}^{-1}(\delta_{G_0})}} f(G')\psi_{G_0}(G')$$

If $G' \in h_{G_0}^{-1}(\delta_{G_0})$, then $d(G', G_0) > r$, and therefore $|f(G') - f(G_0)| < \epsilon$. Hence, we can use $\bar{h}(G') = \sum_{G_0 \in K_0} f(G_0)\psi_{G_0}(G')$ to approximate $f(G')$, because

$$|f(G') - \sum_{\substack{G_0 \in K_0}} f(G_0)\psi_{G_0}(G')| = |f(G') - \sum_{\substack{G_0 \in K_0 \\ G' \in h_{G_0}^{-1}(\delta_{G_0})}} f(G_0)\psi_{G_0}(G')|$$

$$= \sum_{\substack{G_0 \in K_0 \\ G' \in h_{G_0}^{-1}(\delta_{G_0})}} |f(G') - f(G_0)|\psi_{G_0}(G') \qquad (2)$$

$$< \epsilon$$

Finally, we need to show how to approximate $\bar{h}$ with functions from $\mathcal{C}$ augmented with a multi-layer perceptron. We start with $\{h_{G_0}\}_{G_0 \in K} \subseteq \mathcal{C}$, and apply them to the input graph $G'$. Then, for each of $h_{G_0}G'()$ apply an MLP with one hidden layer to obtain $\varphi_{G_0}(G')$, and use one node to store. their sum, $\varphi(G')$. We then use an MLP with one hidden layer to approximate division, obtaining $\psi_{G_0}(G')$. Finally, $\bar{h}(G')$ is approximated by a linear combination of $\{\psi_{G_0}(G')\}_{G_0 \in K}$, since each $f(G_0)$ is a constant.

$\square$

# B    Proofs of Section 4.2

**Theorem 5.** If $\mathcal{C}$ is a class of permutation-invariant functions on $\mathcal{X}^{n \times n}$ and $\mathcal{C}$ is GIso-discriminating, then $\sigma(\mathcal{C}) = \sigma(Q_K)$

*Proof of Theorem 5.* If $\mathcal{C}$ is GIso-discriminating, then given a $G \in \mathcal{X}^{n \times n}, \forall G' \not\simeq G, \exists h_{G'} \in \mathcal{C}$ and $b_{G'} \in \mathbb{R}$ such that $\mathcal{E}(G) = \cap_{G' \not\simeq G} h_{G'}^{-1}(\{b'_G\})$, which is a finite intersection of sets in $\sigma(\mathcal{C})$. Hence, $\mathcal{E}(G) \in \sigma(f_G) \subseteq \sigma(\mathcal{C})$. Therefore, $Q_K \subseteq \sigma(\mathcal{C})$, and hence $\sigma(Q_K) \subseteq \sigma(\mathcal{C})$. Moreover, since $\sigma(g) \subseteq \sigma(Q_K)$ for all $g \in \mathcal{C}$, there is $\sigma(\mathcal{C}) \subseteq \sigma(Q_K)$

$\square$

**Theorem 6.** Let be $\mathcal{C}$ a class of permutation-invariant functions on $\mathcal{X}^{n \times n}$ with $\sigma(\mathcal{C}) = \sigma(Q_K)$. Then $\mathcal{C}$ is GIso-discriminating.

*Proof of Theorem 6.* Suppose not. This implies that $Q_K \subsetneq \sigma(\mathcal{C})$, and hence $\exists \tau = \mathcal{E}(G) \in Q_K$ such that $\tau \notin \sigma(\mathcal{C})$. Note that $\tau$ is an equivalence class of graphs that are isomorphic to each other. Then consider the smallest subset in $\sigma(\mathcal{C})$ that contains $\tau$, defined as $S(\tau) = \bigcap_{\substack{T \in \sigma(\mathcal{C}) \\ \tau \subseteq T}} T$.

Since $K$ is a finite space, $\sigma(\mathcal{C})$ is also finite, and hence this is a finite intersection. Since a sigma-algebra is closed under finite intersection, there is $S(\tau) \in \sigma(C)$. As $\tau \notin \sigma(\mathcal{C})$, we know that $\tau \subsetneq S(\tau)$. Then, $\exists G' \not\cong G$ such that $G' \in S(\tau)$. Then there does not exist any function $h$ in $\mathcal{C}$ such that $h(G) \neq h(G')$, since otherwise the pre-image of some interval in $\mathbb{R}$ under $h$ will intersect with only $\mathcal{E}(G)$ but not $\mathcal{E}(G')$. Contradiction. □

## C   Comparison of the expressive power of families of functions via the sigma-algebra framework

Given two classes of functions $\mathcal{C}_1, \mathcal{C}_2$, such as two classes of GNNs, there are four possibilities regarding their relative representation power, using the language of sigma-algebra developed in the main text:

- $\sigma(\mathcal{C}_1) = \sigma(\mathcal{C}_2)$
- $\sigma(\mathcal{C}_1) \subsetneq \sigma(\mathcal{C}_2)$
- $\sigma(\mathcal{C}_2) \subsetneq \sigma(\mathcal{C}_1)$
- Not comparable / None of the above (i.e., $\sigma(\mathcal{C}_1) \nsubseteq \sigma(\mathcal{C}_2)$ and $\sigma(\mathcal{C}_1) \nsupseteq \sigma(\mathcal{C}_2)$)

In this section we summarize some results from the literature and show partial relationships between different GNNs architectures in terms of their ability to distinguish non-isomorphic graphs (in the context of the sigma-algebra framework introduced in Section 4). For simplicity, in this section we assume that graphs are given by an adjacency matrix (no node or edge features are considered), and the findings are illustrated in Figure 1.

- **sGNN($\mathcal{M}$).** We consider spectral GNNs as the ones used in [5] for community detection. In this context we focus on the simplified version where the GNNs are defined as

$$v^0 = \mathbb{1}_n$$

$$v^{t+1} = \rho \left( \sum_{M \in \mathcal{M}} M v^t \theta_M^t \right) \text{ where } \theta_M^t \in \mathbb{R}^{d_t \times d_{t+1}} \text{ learnable parameters, } v^t \in \mathbb{R}^{n \times d_t}$$

$$\text{output} : \sum_{i=1}^{d_L} v_i^L.$$

  Usually $\mathcal{M}$ is a set of operators related to the graph. In this context we consider $\mathcal{M} = \{I, A\}$ and $\mathcal{M}_{(J)} = \{I, D, A, \min\{A^{2^t}, 1\}, \ t = 2, \ldots\}$. The operators $\min\{A^{2^t}, 1\}$ allow the model to distinguish regular graphs that order 2 G-invariant networks cannot distinguish, such as the Circular Skip Link graphs.

- **Linear Programming (LP).** This is not a GNN but the natural linear programming relaxation for graph isomorphism. Namely given a pair graphs with adjacency matrix $A, B \in \{0, 1\}^{n \times n}$

$$LP(A, B) = \min \|PA - BP\|_1 \text{ subject to } P\mathbb{1}_n = \mathbb{1}_n, \ P^\intercal \mathbb{1}_n = \mathbb{1}_n, \ P \geq 0.$$

  The natural sigma algebra to consider here is $\sigma(\cup_{A \in \mathcal{X}^{n \times n}} \{LP(A, \cdot)\})$. Two graphs are said to be fractionally isomorphic is $LP(A, B) = 0$ (i.e. the LP cannot distinguish them). [24] showed that two graphs are fractionally isomorphic if and only if they cannot be distinguished by 1-WL.

- **Semidefinite Programming (SDP)**. The semidefinite programming relaxation of quadratic assignment from [37] is based on the following observation: $\|PA - BP\|_F^2 = \|PA\|_F^2 + \|BP\|_F^2 - 2\operatorname{trace}(PAP^\top B^\top)$ and $\operatorname{trace}(\operatorname{vec}(P)\operatorname{vec}(P)^\top A \otimes B^\top)$ where $\otimes$ is the Kronecker product operator and vec takes an $n \times n$ matrix and flattens it into an $n^2 \times 1$ vector. The resulting semidefinite relaxation considers the vector $x^\top := [1, \operatorname{vec}(P)^\top]$ and relaxes the rank 1 matrix $xx^\top$ into a positive semidefinite matrix. By including the constraints corresponding to the LP in $xx^\top$ one makes sure that solution of the SDP is always in the feasible set of the LP, therefore the LP is less expressive than the SDP.

- **Sum-of-Squares (SoS) hierarchy**. One can consider the hierarchy of relaxations coming from sum-of-squares (SoS). In the context of graph isomorphism, it is known that graph isomorphism is a hard problem for this hierarchy [23]. In particular the Lasserre/SoS hierarchy requires $2^{\Omega}(n)$ to solve graph isomorphism (in the same sense that $o(n)$-WL fails to solve graph isomorphism [4]).

- **Spectral methods**. If we consider the function that takes a graph and outputs the set of eigenvalues of its adjacency matrix, such function is permutation invariant. A priori one may think that such function, being highly non-linear, is more expressive than any form message passing GNN. In fact, regular graphs are not distinguished by 1-WL or order 2 $G$-invariant networks and may be distinguished by their eigenvalues (like the Circular Skip Link graphs). However, 1-WL and this particular spectral method are not comparable (a simple example is provided in Figure 2 of [24]).

## D Relationship to *Bloom-Reddy and Teh (2019)* [2]

This work [2] provides a nice and general theoretical framework that establishes equivalence between functional and probabilistic perspectives to symmetry via noise outsourcing in both general and particular settings. Our framework belongs to the functional perspective to symmetry (in particular $\mathbb{S}_{n_2}$-invariance), and an extension to the probabilistic perspective with ideas from Bloom-Reddy and Teh would be quite interesting. The concept of orbits also applies in our setting, and the concept of maximal invariants is related to our definition of GIso-discriminating. However, a key distinction is that being a maximal invariant is a property of functions, whereas we define GIso-discriminating to be a property of *classes* of functions. Our definition is arguably more suitable for studying the representation power of different GNN architectures, and moreover makes it possible to relate graph isomorphism testing to function approximation. Furthermore, our theoretical framework described in section 4 focuses on sigma-algebras generated by classes of GNN functions when they are not necessarily GIso-discriminating, allowing us to compare their representation powers to each other, which is another novel contribution.

## E Graph G-invariant Networks with maximum tensor order 2

In this section we prove Theorem 7 that says that graph G-invariant Networks with tensor order 2 cannot distinguish between non-isomorphic regular graphs with the same degree.

### E.1 Defining the order-2 graph $G$-invariant Networks

Here, we state our definition of order-2 Graph $G$-invariant networks based on the $G$-invariant networks defined in [19].

**Notation 1.** *Suppose $A \in \mathbb{R}^{n^k \times a}$ is a tensor containing graph data, where each entry is associated with a $k$-tuple of nodes. Then $\forall \pi \in S_n$, we use $\pi * A$ to denote the $\mathbb{R}^{n^k \times a}$ tensor transformed from $A$ by applying the permutation $\pi$ to the node set. For example, if $A \in \mathbb{R}^{n \times n}$ is a matrix containing edge features (a simple example being the adjacency matrix), then $\pi * A = \pi^\top A \pi$.*

**Definition 5.** *A function $f : \mathbb{R}^{n^k \times a} \to \mathbb{R}^b$ is graph-G-invariant if $\forall A \in \mathbb{R}^{n^k \times a}, \forall \pi \in S_n, f(\pi * A) = f(A)$. A function $f' : \mathbb{R}^{n^k \times a} \to \mathbb{R}^{n^l \times b}$ is graph-G-equivariant if $\forall A \in \mathbb{R}^{n^k \times a}, \forall \pi \in S_n, f'(\pi * A) = \pi * f(A)$. Thus graph-G-invariance is a special case of graph-G-equivariance when $l = 0$.*

**Definition 6.** *A Graph $G$-invariant network is a function $F : \mathbb{R}^{n^{k_0} \times d_0} \to \mathbb{R}$ that can be decomposed in the following way:*

$$F = m \circ h \circ L_T \circ \sigma \circ \cdots \circ \sigma \circ L_1,$$

*where each $L_i$ is a linear graph-$G$-equivariant layer from $\mathbb{R}^{n^{k_{i-1}} \times d_{i-1}}$ to $\mathbb{R}^{n^{k_i} \times d_i}$, $\sigma$ is a pointwise activation function, $h$ is a graph-$G$-invariant layer from $\mathbb{R}^{n^{k_T} \times d_T}$ to $\mathbb{R}$, and $m$ is an MLP.*

By restricting the tensor order to be 2 at each intermediate layer in the definition above, we arrive at the following definition.

**Definition 7.** *An order-2 Graph $G$-invariant network is a function $F : \mathbb{R}^{n \times n \times d_0} \to \mathbb{R}$ that can be decomposed in the following way:*

$$F = m \circ h \circ L_T \circ \sigma \circ \cdots \circ L_1,$$

*where each $L_i$ is a linear graph $G$-equivariant layer from $\mathbb{R}^{n \times n \times d_{i-1}}$ to $\mathbb{R}^{n \times n \times d_i}$, $\sigma$ is a pointwise activation function, $h$ is a graph $G$-invariant layer from $\mathbb{R}^{n \times n \times d_T}$ to $\mathbb{R}$, and $m$ is an MLP.*

We use $A^{(t)}$ to denote the output of the $t$th layer, for $t \in \{1, ..., t\}$, i.e., they are defined recursively by

$$A^{(t+1)} = \sigma(L^{(t)}(A^{(t)}))$$

where $A^{(0)} \in \mathbb{R}^{n \times n \times d_0}$ is the input tensor.

## E.2  Proof of Theorem 7

In the definition above, $d_t$ is the feature dimension in layer $t$, interpreted as the dimension of the hidden state attached to each pair of nodes. For simplicity of notations, in the following proof we assume that $d_t = 1, \forall t = 0, 1, ..., L$, and thus each $A^{(t)}$ is essentially a matrix. The following results can be extended to the cases where $d_t > 1$, by adding more subscripts in the proof.

To prove Theorem 7, we show that if we use the adjacency matrices of two non-isomorphic regular graphs with the same degree as inputs to any order-2 graph $G$-invariant network, the network will return the same output. Notation-wise, given an unweighted graph $G$, let $E \subseteq [n]^2$ be the edge set of $G$, i.e., $(u,v) \in E$ if $u \neq v$ and $G_{uv} = 1$; set $S \subseteq [n]^2$ to be $\{(u,u)\}_{u \in [n]^2}$; and let $N = [n]^2 \setminus (E \cup S)$. Thus, $E \cup N \cup S = [n]^2$.

**Lemma 5.** *Let $G, G'$ be the adjacency matrices of two unweighted regular graphs with the same degree $d$, and let $A^{(t)}, E, N, S$ and $A'^{(t)}, E', N', S'$ be defined as above for $G$ and $G'$, respectively. Then $\forall n \leq L, \exists \xi_1^{(t)}, \xi_2^{(t)}, \xi_3^{(t)} \in \mathbb{R}$ such that $A_{uv}^{(t)} = \xi_1^{(t)} \mathbb{1}_{(u,v) \in E} + \xi_2^{(t)} \mathbb{1}_{(u,v) \in N} + \xi_3^{(t)} \mathbb{1}_{(u,v) \in S}$, and $A_{uv}'^{(t)} = \xi_1^{(t)} \mathbb{1}_{(u,v) \in E'} + \xi_2^{(t)} \mathbb{1}_{(u,v) \in N'} + \xi_3^{(t)} \mathbb{1}_{(u,v) \in S'}$*

*Proof.* We prove this lemma by induction. For $t = 0$, $A^{(0)} = G$ and $A'^{(0)} = G'$. Since the graph is unweighted, $G_{uv} = 1$ if $u \neq v$ and $(u,v) \in E$, and 0 otherwise. Similar is true for $G'$. Therefore, we can set $\xi_1^{(0)} = 1$ and $\xi_2^{(0)} = \xi_3^{(0)} = 0$.

Next, we consider the inductive steps. Assume that the conditions in the lemma are satisfied for layer $t - 1$. To simplify the notation, we use $A, A'$ to stand for $A^{(t-1)}, A'^{(t-1)}$, and we assume to satisfy the inductive hypothesis with $\xi_1, \xi_2$ and $\xi_3$. We thus want to show that if $L$ is any equivariant linear, then $\sigma(L(A)), \sigma(L(A'))$ also satisfies the inductive hypothesis. Also, in the following, we use $p_1, p_2, q_1, q_2$ to refer to nodes, $a, b$ to refer to pairs of nodes, $\lambda$ to refer to any equivalence class of 2-tuples (i.e. pairs) of nodes, and $\mu$ to refer to any equivalence class of 4-tuples of nodes.

$\forall a = (p_1, p_2), b = (q_1, q_2) \in [n]^2$, let $\mathcal{E}(a, b)$ denote the equivalence class of 4-tuples containing $(p_1, p_2, q_1, q_2)$, and let $\mathcal{E}(b)$ represent the equivalence class of 2-tuples containing $(q_1, q_2)$. Two 4-tuples $(u, v, w, x), (u', v', w', x')$ are considered equivalent if $\exists \pi \in S_n$ such that $\pi(u) = u', \pi(v) = v', \pi(w) = w', \pi(x) = x'$. Similarly is equivalence between 2-tuples defined. By equation 9(b) in [18], using the notations of $T, B, C, w, \beta$ defined there, $L$ is described by, given $A$ as an input as $b$ as

Figure 4: $m_E(E, \mathcal{E}(1,2,3,4))$, $m_E(E, \mathcal{E}(1,2,3,2))$, $m_E(E, \mathcal{E}(1,2,3,1))$, $m_E(E, \mathcal{E}(1,2,2,3))$ and $m_E(E, \mathcal{E}(1,2,1,3))$ of $G_{8,2}$ and $G_{8,3}$. In either graph, twice the total number of black edges equal $m_E(E, \mathcal{E}(1,2,3,4)) = 18$ (it is twice because each undirected edge corrspond to two pairs $(p_1, p_2)$ and $(p_2, p_1)$, which combined with $(q_1, q_2)$ both belongs to $\mathcal{E}(1,2,3,4)$); the total number of of red edges, 3, equals both $m_E(E, \mathcal{E}(1,2,2,3))$ and $m_E(E, \mathcal{E}(1,2,1,3))$; the total number of green edges, also 3, equals both $m_E(E, \mathcal{E}(1,2,3,2))$, $m_E(E, \mathcal{E}(1,2,3,1))$.

the subscript index on the output,

$$
\begin{aligned}
L(A)_b &= \sum_{a=(p_1,p_2)=(1,1)}^{(n,n)} T_{a,b} A_a + Y_b \\
&= \sum_{a,\mu} w_\mu B_{a,b}^\mu A_a + \sum_\lambda \beta_\lambda C_b^\lambda \\
&= \sum_\mu \Big( \sum_{\substack{a \in [n]^2 \\ (a,b) \in \mu}} A_a \Big) w_\mu + \beta_{\mathcal{E}(b)}
\end{aligned}
\tag{3}
$$

First, let

$$
S_\mu^b = \sum_{\substack{a \in [n]^2 \\ (a,b) \in \mu}} A_a
$$

By the inductive hypothesis,

$$
\begin{aligned}
S_\mu^b &= \sum_{\substack{a \in [n]^2 \\ (a,b) \in \mu \\ a \in E}} A_a + \sum_{\substack{a \in [n]^2 \\ (a,b) \in \mu \\ a \in N}} A_a + \sum_{\substack{a \in [n]^2 \\ (a,b) \in \mu \\ a \in S}} A_a \\
&= \sum_{\substack{a \in [n]^2 \\ (a,b) \in \mu \\ a \in E}} \xi_1 + \sum_{\substack{a \in [n]^2 \\ (a,b) \in \mu \\ a \in N}} \xi_2 + \sum_{\substack{a \in [n]^2 \\ (a,b) \in \mu \\ a \in S}} \xi_3 \\
&= m_E(b,\mu)\xi_1 + m_N(b,\mu)\xi_2 + m_S(b,\mu)\xi_3
\end{aligned}
\tag{4}
$$

where $m_E(b,\mu)$ is defined as the total number of distinct $a \in [n]^2$ that satisfies $(a,b) \in \mu$ and $a \in E$, and similarly for $m_N(b,\mu)$ and $m_S(b,\mu)$. Formally, for example, $m_E(b,\mu) = card\{a \in [n]^2 : (a,b) \in \mu, a \in E\}$.

Since $E \cup N \cup S = [n]^2$, $b$ belongs to one of $E, N$ and $S$. Thus, let $\tau(b) = E$ if $b \in E$, $\tau(b) = N$ if $b \in N$ and $\tau(b) = S$ if $b \in S$. It turns out that if $A$ is the adjacency matrix of a undirected regular graph with degree $d$, then $m_E(b,\mu), m_N(b,\mu), m_S(b,\mu)$ can be instead written (with an abuse of notation) as $m_E(\tau(b),\mu), m_N(\tau(b),\mu), m_S(\tau(b),\mu)$, meaning that for a fixed $\mu$, the values of $m_E, m_N$ and $m_S$ only depend on which of the three sets ($E, N$ or $S$) $b$ is in, and changing $b$ to a different member in the set $\tau(b)$ won't change the three numbers. In fact, for each $\tau(b)$ and $\mu$, the three numbers can be computed as functions of $n$ and $d$ using simple combinatorics, and their values are seen in the three tables 3, 4 and 5. An illustration of these numbers is given in Figure E.2.

Therefore, we have $L(A)_b = \sum_\mu w_\mu(m_E(\tau(b),\mu) + m_N(\tau(b),\mu) + m_S(\tau(b),\mu)) + \beta_{\mathcal{E}(b)}$. Moreover, notice that $\tau(b)$ determines $\mathcal{E}(b)$: if $\tau(b) = E$ or $N$, then $\mathcal{E}(b) = \mathcal{E}(1,2)$; if $\tau(b) = S$, then

| $\mu$ | $m_E(E,\mu)$ | $m_E(N,\mu)$ | $m_E(S,\mu)$ |
|---|---|---|---|
| $(1,2,3,4)$ | $(n-4)d+2$ | $(n-4)d$ | $0$ |
| $(1,1,2,3)$ | $0$ | $0$ | $0$ |
| $(1,2,2,3)$ | $d-1$ | $d$ | $0$ |
| $(1,2,1,3)$ | $d-1$ | $d$ | $0$ |
| $(1,2,3,2)$ | $d-1$ | $d$ | $0$ |
| $(1,2,3,1)$ | $d-1$ | $d$ | $0$ |
| $(1,1,1,2)$ | $0$ | $0$ | $0$ |
| $(1,1,2,1)$ | $0$ | $0$ | $0$ |
| $(1,2,1,2)$ | $1$ | $0$ | $0$ |
| $(1,2,2,1)$ | $1$ | $0$ | $0$ |
| $(1,2,3,3)$ | $0$ | $0$ | $(n-2)d$ |
| $(1,1,2,2)$ | $0$ | $0$ | $0$ |
| $(1,2,2,2)$ | $0$ | $0$ | $d$ |
| $(1,2,1,1)$ | $0$ | $0$ | $d$ |
| $(1,1,1,1)$ | $0$ | $0$ | $0$ |
| Total | $nd$ | $nd$ | $nd$ |

Table 3: $m_E$

| $\mu$ | $m_N(E,\mu)$ | $m_N(N,\mu)$ | $m_N(S,\mu)$ |
|---|---|---|---|
| $(1,2,3,4)$ | $(n-4)(n-d-1)$ | $(n-4)(n-d-1)+2$ | $0$ |
| $(1,1,2,3)$ | $0$ | $0$ | $0$ |
| $(1,2,2,3)$ | $n-d-1$ | $n-d-2$ | $0$ |
| $(1,2,1,3)$ | $n-d-1$ | $n-d-2$ | $0$ |
| $(1,2,3,2)$ | $n-d-1$ | $n-d-2$ | $0$ |
| $(1,2,3,1)$ | $n-d-1$ | $n-d-2$ | $0$ |
| $(1,1,1,2)$ | $0$ | $0$ | $0$ |
| $(1,1,2,1)$ | $0$ | $0$ | $0$ |
| $(1,2,1,2)$ | $0$ | $1$ | $0$ |
| $(1,2,2,1)$ | $0$ | $1$ | $0$ |
| $(1,2,3,3)$ | $0$ | $0$ | $(n-2)(n-d-1)$ |
| $(1,1,2,2)$ | $0$ | $0$ | $0$ |
| $(1,2,2,2)$ | $0$ | $0$ | $n-d-1$ |
| $(1,2,1,1)$ | $0$ | $0$ | $n-d-1$ |
| $(1,1,1,1)$ | $0$ | $0$ | $0$ |
| Total | $n(n-d-1)$ | $n(n-d-1)$ | $n(n-d-1)$ |

Table 4: $m_N$

$\mathcal{E}(b) = \mathcal{E}(1,1)$. Hence, we can write $\beta_{\tau(b)}$ instead of $\beta_{\mathcal{E}(b)}$ without loss of generality. Then in particular, this means that $L(A)_b = L(A)_{b'}$ if $\tau(b) = \tau(b')$. Therefore, $L(A)_b = \overline{\xi}_1 \mathbb{1}_{b\in E} + \overline{\xi}_2 \mathbb{1}_{b\in N} + \overline{\xi}_3 \mathbb{1}_{b\in S}$, where $\overline{\xi}_1 = \sum_\mu w_\mu(m_E(E,\mu)+m_N(E,\mu)+m_S(E,\mu))+\beta_E, \overline{\xi}_2 = \sum_\mu w_\mu(m_E(N,\mu)+ m_N(N,\mu) + m_S(N,\mu)) + \beta_N$, and $\overline{\xi}_3 = \sum_\mu w_\mu(m_E(S,\mu) + m_N(S,\mu) + m_S(S,\mu)) + \beta_S$.

Similarly, $L(A')_b = \overline{\xi'}_1 \mathbb{1}_{b\in E'} + \overline{\xi'}_2 \mathbb{1}_{b\in N'} + \overline{\xi'}_3 \mathbb{1}_{b\in S'}$. But importantly, $\forall$ equivalence class of 4-tuples, $\mu$, and $\forall \lambda_1, \lambda_2 \in \{E, N, S\}, m_{\lambda_1}(\lambda_2,\mu) = m'_{\lambda_1}(\lambda_2,\mu)$, as both of them can be obtained from the same entry of the same table. Therefore, $\overline{\xi}_1 = \overline{\xi'}_1, \overline{\xi}_2 = \overline{\xi'}_2, \overline{\xi}_3 = \overline{\xi'}_3$.

Finally, let $\xi^*_1 = \sigma(\overline{\xi}_1), \xi^*_2 = \sigma(\overline{\xi}_2)$, and $\xi^*_3 = \sigma(\overline{\xi}_3)$. Then, there is $\sigma(L(A))_b = \xi^*_1 \mathbb{1}_{b\in E} + \xi^*_2 \mathbb{1}_{b\in N} + \xi^*_3 \mathbb{1}_{b\in S}$, and $\sigma(L(A'))_b = \xi^*_1 \mathbb{1}_{b\in E'} + \xi^*_2 \mathbb{1}_{b\in N'} + \xi^*_3 \mathbb{1}_{b\in S'}$, as desired. $\square$

Since $h$ is an invariant function, $h$ acting on $A^{(L)}$ essentially computes the sum of all the diagonal terms (i.e., for $b \in S$) and the sum of all the off-diagonal terms (i.e., for $b \in E\cup N$) of $A^{(L)}$ separately and then adds the two sums with two weights. If $G, G'$ are regular graphs with the same degree, then $|E| = |E'|, |S| = |S'|$ and $|N| = |N'|$. Therefore, by the lemma, there is $h(A^{(L)}) = h(A'^{(L)})$, and as a consequence $m(h(A^{(L)})) = m(h(A'^{(L)}))$.

| $\mu$ | $m_S(E,\mu)$ | $m_S(N,\mu)$ | $m_S(S,\mu)$ |
|---|---|---|---|
| $(1, 2, 3, 4)$ | 0 | 0 | 0 |
| $(1, 1, 2, 3)$ | $n-2$ | $n-2$ | 0 |
| $(1, 2, 2, 3)$ | 0 | 0 | 0 |
| $(1, 2, 1, 3)$ | 0 | 0 | 0 |
| $(1, 2, 3, 2)$ | 0 | 0 | 0 |
| $(1, 2, 3, 1)$ | 0 | 0 | 0 |
| $(1, 1, 1, 2)$ | 1 | 1 | 0 |
| $(1, 1, 2, 1)$ | 1 | 1 | 0 |
| $(1, 2, 1, 2)$ | 0 | 0 | 0 |
| $(1, 2, 2, 1)$ | 0 | 0 | 0 |
| $(1, 2, 3, 3)$ | 0 | 0 | 0 |
| $(1, 1, 2, 2)$ | 0 | 0 | $n-1$ |
| $(1, 2, 2, 2)$ | 0 | 0 | 0 |
| $(1, 2, 1, 1)$ | 0 | 0 | 0 |
| $(1, 1, 1, 1)$ | 0 | 0 | 1 |
| Total | $n$ | $n$ | $n$ |

Table 5: $m_S$

# F   Specific GNN Architectures

In section 6, we show experiments on synthetic and real datasets with several related architectures. Here are some explanations for them.

- **sGNN-$i$**: sGNNs with operators from family $\{I, D, \min(A^{2^0}, 1), \ldots, \min(A^{2^{i-1}}, 1)\}, i \in \{1, 2, 5\}$. In our experiments, the $sGNN$ models have 5 layers and hidden layer dimension (i.e. $d^k$) 64. They are trained using the Adam [14] optimizer with learning rate 0.01.

- **LGNN**: Line Graph Neural Networks proposed by [5]. In our experiments, the $LGNN$ models have 5 layers and hidden layer dimension (i.e. $d^k$) 64. They are trained using the Adam [14] optimizer with learning rate 0.01.

- **GIN**: Graph Isomorphism Network by [32]. We took their performance results on the IMDB datasets reported in [32], and their performance results on the Circular Skip Link graphs experiments reported in [21].

- **RP-GIN**: Graph Isomorphism Network combined with Relational Pooling by [21]. We took the results reported in [21] for the Circular Skip Link graphs experiment.

- **Order-2 Graph $G$-invariant Networks**: $G$-invariant networks based on [18] and [19], as implemented in https://github.com/Haggaim/InvariantGraphNetworks.

- **Ring-GNN**: The definition is given in the main text. For experiments on IMDB datasets, the "Ring-GNN" model has the same depth and widths of hidden layers as the order-2 Graph $G$-invariant Networks reported in [18]. The "Ring-GNN w/ degree" model has 2 Ring-GNN layers with 64 hidden units in each, followed by a jump knowledge network [33], which is then followed by a fully-connected layer with 32 hidden units. Each $k_1^{(t)}$ is initialized independently under $\mathcal{N}(0, 1)$, and each $k_2^{(t)}$ is initialized independently under $\mathcal{N}(0, 0.01)$. They are trained using the Adam [14] optimizer with learning rate 0.00001 for 350 epochs. The initialization of $k_2^{(t)}$ and the learning rate were manually tuned, following the heuristic that Ring-GNN reduces to order-2 Graph $G$-invariant Networks when $k_2^{(t)} = 0$, and that since Ring-GNN added more operators, a smaller learning rate is likely more appropriate. For the other real-world datasets, models are trained via Adam [14], with learning rate of 0.001 for 350 epochs. The model has 1 Ring-GNN layer for MUTAG, 2 Ring-GNN layers for PROTEINS and PTC, and 3 Ring-GNN layers for COLLAB. Each of these layers has 64 hidden units. The Ring-GNN layer(s) is followed by a jump knowledge network [33], which is then followed by a fully-connected layer with 32 hidden units. $k_1^{(t)}$ is initialized as 1, while $k_2^{(t)}$ is initialized with $\{0.5/n, 1.0/n\}$ where $n$ is the average number of nodes per graph in each dataset.

For the experiments with Circular Skip Links graphs, each model is trained and evaluated using 5-fold cross-validation. For Ring-GNN, in particular, we performed training + cross-validation 20 times with different random seeds.