[Reviews · NeurIPS 2019]

Reviewer 1



The paper targets the problem of measuring the representation power of Graph neural networks (GNNs), an interesting and important topic, that has become popular recently (partially due to two prominent works (Xu et al. ICLR 2019, Morris et al. AAAI 2019)). There are three main contributions: 1. Establishing the equivalence between two methods for measuring GNN representation power: (i) their ability to approximate permutation invariant functions (ii) their ability to distinguish non-isomorphic graphs. Although not very surprising, this is a nice observation. 2. The authors further suggest reformulating these GNN representation power measures in the language of sigma algebras: Given a class of GNN models, one can naturally associate a sigma algebra (defined on the domain of these models). The authors show that these sigma algebras are an equivalent way to measure representation power of GNNs, for instance, the inclusion of sigma algebras originating from two models is equivalent to saying one model is more powerful than the other. This is a potentially useful observation. I would see it as a stronger contribution if the authors could identify a case in which using the sigma algebra formulation is useful (e.g., easier than using the other representation power measures.) 3. The authors propose a variant of the 2-Graph G-invariant neural networks (Maron et al., ICLR 2019). This variant is shown to be more powerful than many popular graph learning models such as message-passing networks and the 2-G-invariant networks mentioned above. I think this is a very good direction although the experimental section lacks proper comparison on multiple datasets. Here are a few additional comments/reservations: 1. Clarity: in general the paper is well structured but there are several places that can be improved. For example definition 3 is long and hard to parse, the high-level idea behind Lemmata 3-4. Similarly, definition 5 is too long. Illustrations might help. 2. Section 4: can the authors discuss the case of infinite K? 3. I am missing a more thorough evaluation of the Ring-GNN. 4. Figure1: can the authors clearly write how each arrow is obtained? 5. The long proofs in the appendix are hard to follow. Can the authors try to make them more accessible? (maybe by using illustrations)? In general, this paper presents two potentially useful theoretical observations on measuring the expressive power of GNNs and proposes a strong GNN variant that looks promising. -------------------------------------------------------------------------- Post Rebuttal -------------------------------------------------------------------------- After reading all other reviews and author response I remain on the positive side. The new experimental results strengthen the paper. The SVD part, as other reviewers pointed out, should be changed. I think that the authors should compare to other eigenvalue based methods, or provide results without the SVD part.

Reviewer 2



originality: O1. The paper is based on G-invariant networks but uses an interesting idea to improve its power. O2. The use of measure theory is not new (e.g., Bloem-Reddy and Teh [2]) but the connections to function approximation were interesting. quality: Q1. I like the paper very much. It is great to see the works on GNNs adding more formalism. Ring-GNN is a simple but interesting idea, add the powers of A to the G-invariant network representation. Q1.1 I really liked the use of the CSL task, it provides a clean way to show progress in the representation power. Q2. [IMPORTANT] Adding SVD to Ring-GNN feels like cheating. What if k-GNN and RP-GIN methods were also given the eigenvalues [say, concatenated with their embeddings passing through an MLP]? The eigenvalues of the 10 CSL classes are likely different, even a simple MLP may be able to distinguish the classes without the GNN. Either show this is not the case, or remove the SVD approach. Q3. The sigma-algebra formalism has many parallels to that of Bloem-Reddy and Teh [2], but I like the former better for the added probabilistic interpretation. Noise outsourcing should allow this paper to connect the sigma-algebra formalism with representation learning and generative models. I think tightening the connection with Bloem-Reddy and Teh would strengthen this paper. Also, introducing the concept of orbit would be useful in the formalism. Q4. One of the main competitors on experiment 6.1 (RP-GIN) should be described in the introduction with GIN. Q5. When introducing CSL graphs (first mention of Figure 5), it would be useful to cite Murphy et al. [19] where they are described in more detail, otherwise it looks like something that the reader should just know. clarity: C1- I really enjoyed reading the appendix, thanks for all the effort that went into it! (illustrations, examples, detailed descriptions, extensions are all great) C2- The statement "Such a dependence on the graph size was been theoretically overcame by the very recent work [13]" right after "showed the universal approximation of G-invariant networks, constructed based on the linear invariant and equivariant layers studied in [16], if the order of the tensor involved in the networks can grow as the graph gets larger" is strangely worded , as it seem to imply [13] was abe to overcome the large order tensor that universality needs (which is contradicted after the comma in the next page). C3- " [19] proposes relational pooling", ". [2] studies the", ." [16] studies the spaces" => it is odd to start a sentence with a number. Please add 1st author's names when rather than using numbers as nouns. C4 - Section 6.1 does not comment on the Ring-GNN-SVD. Overall, I feel the SVD addition to Ring-GNN without adding the eigenvalues to other methods is cheating. typo: sigmas-algebras => sigma-algebras

Reviewer 3



Pros: 1, The theoretical analysis on the connection between the universal approximation of permutation invariant functions and the capacity limitation of GNNs is fundamental. 2, The introduction of sigma-algebra formulation of expressiveness is very novel and interesting. 3, The synthetic experiments on circular skip links graphs are very impressive and interesting. Cons: 1, Although I like the theoretical results on the universal approximation and limitations of GNNs, it is a bit disappointing that the newly proposed Ring-GNN does not have any guarantee on the universal approximation, if I understood correctly. 2, To me, it seems that the reason why Ring GNN outperforms other models in the Circular Skip Link graphs is that the construction of the adjacency matrix at different layers uses graph information at different scales. For example, the J + 1 layer adjacency matrix is min(A^2^J, 1). On this perspective, the discussion and comparison with [1] is necessary to me since you can easily treat the exponent of adjacency matrix as a hyperparameter and learn spectral filters in [1]. Moreover, the full SVD is avoided in [1] which breaks the complexity down to O(kN^2) where k is the number of top eigenvalues you want. 3, Details of the model are sparse. Since the space is limited, I suggest authors to trim some of the remarks and lemmas which does not directly help understanding the main results. Instead, you can use the space to properly introduce the G-invariant network which will significantly help readers understand the model in section 5. For example, in definition 5, what is the function rho used in Ring GNN? 4, The computational complexity of Ring GNN is quite high which may be a bottleneck for practical use. 5, The performance of the proposed Ring-GNN on real IMDB datasets is considerably worse than GIN and other baselines. Do you have any idea why this is the case? Also, please provide the reference for the IMDB datasets in the text. 6, The problem setting of the theoretical analysis is different from the practical setting where node feature is presented. Specifically, a graph neural network is not a function which maps an n by n adjacency matrix to scalar but a function which maps a tuple of an n by n adjacency matrix and an n by d node feature to scalar/vector. I wonder how the analysis applies to such a setting. [1] Liao, R., Zhao, Z., Urtasun, R. and Zemel, R.S., 2019. Lanczosnet: Multi-scale deep graph convolutional networks. arXiv preprint arXiv:1901.01484. ============================================================= Thanks for the response! The clarification on Ring-GNN and node feature helps me understand the contribution better. However, I disagree about the comparison with LanczosNet. First, the randomized starting vector would not be a problem if you just fix it for every run which does not hurt the theoretical guarantee of Lanczos algorithm in general (smart choice of starting vector leads to better convergence). Second, the functional form of min(A^2^J, 1) will not make a big difference since if the learnable spectral filter (essentially a MLP) in LanczosNet takes the A^2^J as input, min( , 1) can be approximated by the filter in an arbitrarily accurate manner. At last, your example of "hexagon” and “2 disconnected triangles" can be distinguished by LanczosNet. The reason is the eigenvalues/spectrum of the adjacency matrices are different. One is [-2, -1, 2, -1, 1, 1] and the other is [-1, 2, -1, -1, 2, -1]. LanczosNet not only takes the node feature and propagates but also takes the spectrum as input and extract feature from the spectrum which means the final representation of these graphs are different. Overall, lacking a comparison with GNNs which also leverage the eigenvalues significantly degrades the contribution on the proposed algorithm. On the other hand, if you can remove the SVD part, i.e., every GNNs do not use eigenvalues explicitly, then the comparison seems more fair and convincing. Therefore, I would like to keep my original score.

[Author Response · NeurIPS 2019]

Many thanks to all three reviewers for appreciating our work and providing helpful comments and questions.

**To Reviewer 1**

1. *Why is the sigma-algebra formulation useful*: It allows us to compare the representation power of different classes of GNNs succinctly, precisely by comparing the $\sigma$-algebras generated by them. For example, if the $\sigma$-algebra generated by the class of type A GNNs is a sub-$\sigma$-algebra of that of type B GNNs, then we know that A is no more powerful than B.

2. *Sigma-algebra in the case of infinite feature space*: An extension to countably infinite space is straightforward, while for uncountable space we would need more technical results on $\sigma$-algebras. We therefore leave it for future efforts.

3. *Figure 1 & proofs:* The inclusions in Fig. 1 have been establish in the literature, which we referenced in Appendix C. We will add the references per arrow in the caption. We will also add more explanations and illustrations to the proofs.

4. *More thorough experiments*: We extended the numerical results to other standard datasets: collab, mutag, proteins and ptc, and obtained competitive performance as shown below. Moreover, we found that in the GIN paper's experiments, node degrees were added as node features for the social network datasets, which we did not add in the experiment reported in our paper. After adding them, Ring-GNN's performance on IMDB datasets also improved.

|  | imdbb | imdbm | collab | mutag | ptc | proteins |
|---|---|---|---|---|---|---|
| Ring-GNN | 73.3±4.9 | 51.3±4.24 | 80.12±1.4 | 86.8±6.4 | 65.7±7.13 | 75.65±2.93 |
| GIN | 75.1±5.1 | 52.3±2.8 | 80.2±1.9 | 89.4±5.6 | 64.6±7.0 | 76.2±2.8 |
| G-invariant | 72.0±5.54 | 48.73±3.41 | 78.36±2.47 | 84.61±10 | 59.47±7.3 | 76.58±5.49 |

**To Reviewer 2**

1. *Adding SVD to Ring-GNN is unfair*: Indeed, SVD is quite powerful. We added it to the Ring-GNN model to produce a theoretically very expressive object without going to higher order tensors. However, the fact that the Ring-GNNs are strictly more expressive than order-2 G-invariant networks and spectral GNNs do not rely on adding the SVD. Moreover, the results in the table above show that Ring-GNN *without* SVD can achieve competitive performance on real datasets.

2. *Relation to Bloom-Reddy and Teh*: Their work provides a very nice and general theoretical framework that establishes equivalence between functional and probabilistic perspectives to symmetry via noise outsourcing in both general and particular settings. Our framework belongs to the functional perspective to symmetry (in particular $\mathbb{S}_{n_2}$-invariance), and an extension to the probabilistic perspective with ideas from Bloom-Reddy and Teh would be quite interesting. The concept of orbits also applies in our setting, and the concept of maximal invariants is related to our definition of GIso-discriminating. However, a key distinction is that being a maximal invariant is a property of functions, whereas we define GIso-discriminating to be a property of *classes* of functions. Our definition is arguably more suitable for studying the representation power of different GNN architectures, and moreover makes it possible to relate graph isomorphism testing to function approximation. Furthermore, our theoretical framework described in section 4 focuses on sigma-algebras generated by classes of GNN functions when they are not necessarily GIso-discriminating, allowing us to compare their representation powers to each other, which is another novel contribution.

**To Reviewer 3**

1. *Ring-GNN not guaranteed to be universal*: It is correct that Ring-GNN is not guaranteed to be universal. In fact we don't expect it to be, given that arbitrarily high order tensors are needed to universally approximate [13,17]. However, it is provably more powerful than order-2 G-invariant graph networks, to which Ring-GNN is an extension.

2. *Comparison with LanczosNet*: Indeed, LanczosNet is an interesting model that captures multi-scale information on graphs, which we will add to the references. But firstly, as the initial vector for the Lanczos algorithm is chosen at random, the output of the network is not invariant/equivariant, therefore it's a priori not suitable for graph isomorphism testing. Secondly, as the reviewer pointed out, Ring-GNN is able to express matrices of the form of $min(A^{2^J}, 1)$, which contains some information not present in $A^{2^J}$. For example, consider the two following graphs with 6 nodes and identical node features: G1 is a "hexagon" and G2 is two disconnected "triangles". If we propagate the node features under the adjacency or Laplacian, identical results would be obtained on the two graphs, regardless of the number of iterations. In the language of the LanczosNet paper, the two graphs have identical Krylov subspaces, and hence we suspect that LanczosNet may have trouble telling them apart. On the other hand, propagating the node features under $min(A^2, 1)$ leads to differences between the two graphs, making them distinguishable by Ring-GNNs.

3. *High computational complexity of Ring-GNN*: One computational bottleneck of Ring-GNN is the SVD, which is of theoretical interest but in practice it can be removed with comparative empirical performance (see table above).

4. *Theoretical analysis does not include node features*: Our analyses do apply to the case where nodes have features actually, though admittedly the notations might be slightly misleading. In our setup, $G \in \mathcal{X}^{n \times n}$ contains both node and edge features: the diagonal entries correspond to node features, and the off-diagonal ones correspond to edge features. If node features are $d$ dimensional vectors, we can have $\mathcal{X}$ (compact) $\subseteq \mathbb{R}^d$, and the theoretical analyses still apply.

5. *Why Ring-GNN performs worse than GIN on IMDB datasets*: On one hand, we note that experimental performance does not strictly correlate with representation power. On the other, in our latest results, Ring-GNN outperformed GIN on the ptc dataset and had improved accuracy on the IMDB datasets, as explained in the "To Reviewer 1" section above.

[Meta-Review · NeurIPS 2019]

This paper leverages the graph isomorphism problem to study the expressive power of GNNs. In addition, a measure of expressiveness is formalized using sigma-algebras and the authors propose a novel variant of GNN, RING-GNN, that is evaluated in an experimental study where it shows competitive results. The reviewers agree that this is a nice contribution, the theoretical results are interesting (though somehow expected) and that the proposed extension of G-invariant networks is relevant. However, all reviewers agree that the experimental comparison with RING-GNN-SVD is unfair and MUST BE REMOVED in a published version of the paper (that is removing the last line from table 1). One of the reviewer also note that a comparison with LanczosNet should be included (though the lack of comparison is not ground for rejection).